# Evaluation of Calibration Performance of a Low-cost Particulate Matter Sensor Using Collocated and Distant NO$_2$

[1], Kabseok Ko[1], Seokheon Cho[2], and Ramesh R. Rao[2]

[1]Department of Electronics Engineering, Kangwon National University, Chuncheon, 24341, Korea
[2]Qualcomm Institute, University of California, San Diego (UCSD), La Jolla, CA, 92093, USA

**Abstract.** Low-cost optical particle sensors have the potential to supplement existing particulate matter (PM) monitoring systems to provide high spatial and temporal resolution. However, low-cost PM sensors have often shown questionable performance under various ambient conditions. Temperature, relative humidity (RH), and particle composition have been identified as factors that directly affect the performance of low-cost PM sensors. This study investigated if NO$_2$, which creates PM$_{2.5}$ by chemical reactions in the atmosphere, can be used to improve the calibration performance of low-cost PM$_{2.5}$ sensors. To this end, we evaluated the PurpleAir PA-II, called PA-II, a popular air monitoring system that utilizes two low-cost PM sensors that is frequently deployed near air quality monitoring sites of the Environmental Protection Agency (EPA). We selected a single location where 14 PA-II units have operated for more than two years since July 2017. Based on the operating periods of the PA-II units, we then chose the period of Jan. 2018 to Dec. 2019 for study. Among the 14 units, a single unit containing more than 23 months of measurement data with a high correlation between the unit's two PMS sensors was selected for analysis. Daily and hourly PM$_{2.5}$ measurement data from the PA-II unit and a BAM 1020 instrument, respectively, were compared using the federal reference method (FRM), and a per-month analysis was conducted against the BAM-1020 using hourly PM$_{2.5}$ data. In the per-month analysis, three key features, temperature, relative humidity (RH), and NO$_2$, were considered. The NO$_2$, called collocated NO$_2$, was collected from the reliable instrument collocated with the PA-II unit. The per-month analysis showed the PA-II unit had a good correlation (coefficient of determination, $R^2 > 0.819$) with the BAM-1020 during the months of Nov., Dec., and Jan. in both 2018 and 2019, but their correlation intensity was moderate during other months, such as July and Sep. 2018, and Aug., Sep., and Oct. 2019. NO$_2$ was shown to be a key factor in increasing the value of $R^2$ in the months when moderate correlation based on only PM$_{2.5}$ was achieved. This study calibrated a PA-II unit using multiple linear regression (MLR) and random forest (RF) methods based on the same three features used in the analysis studies as well as their multiplicative terms. The addition of NO$_2$ had a much larger effect than that of RH when both PM$_{2.5}$ and temperature were considered for calibration in both models. When NO$_2$, temperature, and relative humidity were considered, the MLR method achieved similar calibration performance to the RF method. In addressing the feasibility of utilizing distant NO$_2$ measurements for calibration in lieu of collocated data, the study highlights the effectiveness of distant NO$_2$ when correlated strongly with collocated measurements. This finding offers a practical solution for situations where obtaining collocated NO$_2$ data proves challenging or costly. We assessed the performance of different PA-II units to determine their efficacy. Our investigation reveals a significant enhancement in calibration performance across different PA-II units upon integrating NO$_2$. Importantly, this improvement remains consistent even when employing models trained with different PA-II units within the same location. Overall, this in-

vestigation emphasizes the significance of NO$_2$ in improving calibration for low-cost PM$_{2.5}$ sensors and presents insights into leveraging distant NO$_2$ measurements as a viable alternative for calibration in the absence of collocated data.

## 1  Introduction

Recently, attention has been paid to particulate matter (PM), which not only has adverse effects on visibility but also can impact human health by contributing to conditions such as cardiovascular disease, asthma, and lung cancer (Liu et al., 2018, 2013). PM that is less than 2.5 µm in diameter, referred to as PM$_{2.5}$, can penetrate the lungs and may thus increase the risk to human health. Globally, the estimated number of adult deaths attributable to PM$_{2.5}$ exposure is over 0.67, 1.6, and 2.1 million for lung cancer,

cardiopulmonary disease, and all causes, respectively (Evans et al., 2013). To minimize the harmful effects, many countries regulate daily and annual PM$_{2.5}$ concentrations by monitoring PM$_{2.5}$ levels at air quality monitoring stations. The monitoring stations use instruments based on Federal Reference Methods (FRMs) or Federal Equivalent Methods (FEMs), which promote high precision and accuracy. The U.S. Environmental Protection Agency (EPA) approves both FRMs and FEMs as official designations for measuring ambient concentrations. Furthermore, the U.S. EPA carries out various cooperative programs,

including those on ambient monitoring methods and technologies, with many other countries in the world. These instruments can provide high-quality measurements of PM$_{2.5}$ concentrations at the installed locations and nearby surroundings. However, these instruments are sparsely distributed due to the high cost of the equipment (ten thousand to tens of thousands of US dollars), so they cannot provide spatial variability. In other words, traditional monitoring stations frequently provide air quality data with poor spatio-temporal resolution, due to the limited number of high quality instruments.

As a cost-effective approach for a dense monitoring network, many stakeholders and researchers have turned to low-cost PM sensors that use a light scattering technique for measurement. In addition to low cost, these sensors have the advantages of low energy consumption and high sampling frequency, and they are easy to deploy and operate compared to traditional monitoring networks. Thus, low-cost PM sensors have been deployed in several communities to measure and report local air quality information (Jiao et al., 2016; PurpleAir, 2018).

However, low-cost PM sensors are not suitable for regulatory purposes because the data reported can be questionable in terms of accuracy, precision, and reliability. In worst-case scenarios, low-cost sensors report no meaningful data at all. Because manufacturers provide limited information on sensors' performance, some studies have been conducted to evaluate the performance of a variety of low-cost sensor models by comparing them with high-cost instruments in laboratory and outdoor ambient environments (Alvarado et al., 2015; Johnson et al., 2018; Wang et al., 2015; Holstius et al., 2014; Austin et al., 2015;

Gao et al., 2015; Kelly et al., 2017; Mukherjee et al., 2017; Sousan et al., 2016; Feinberg et al., 2018; Crilley et al., 2018; Badura et al., 2018; Budde et al., 2018; Liu et al., 2019; Cavaliere et al., 2015; Kelly et al., 2017; Zheng et al., 2018). Most sensors showed good performance under laboratory tests where the sensors measured, known concentrations of particles, such as polystyrene latex, in a chamber. On the other hand, under ambient conditions, the performance of low cost sensors varied depending on the sensor model and its deployed location. Some PM sensor units have inconsistent precision between units of

the same model (Feenstra et al., 2019; Feinberg et al., 2018), while other PM monitors, including the PurpleAir PA-II, have

shown good precision (Barkjohn et al., 2020; Pawar and Sinha, 2020; Mailings et al., 2020). Field evaluations of PurpleAir PA-II units collocated with FEM instruments for approximately two months shown good correlation with the FEM instruments (SCAQMD, 2017c). Furthermore, it was shown that PMS5003 sensors, which are used in PurpleAir PA-II monitors, have a good correlation with the FEM monitors (Kelly et al., 2017; Sayahi et al., 2019). However, the sensors still require calibration for better performance before use in ambient conditions.

Several studies have developed calibration models for low-cost PM sensors based on the following approaches: simple linear regression (Zheng et al., 2018), multiple linear regression (Zimmerman et al., 2018), random forest (Zimmerman et al., 2018), and neural networks (Si et al., 2020). Moreover, to improve calibration performance, several studies have identified other factors in addition to $PM_{2.5}$ concentration that can affect the performance of low-cost sensors. These typical factors include temperature, relative humidity, and particle properties (composition and size distribution) (Holstius et al., 2014; Gao et al., 2015; Kelly et al., 2017). In particular, some low-cost PM sensors have been shown to excessively overestimate $PM_{2.5}$ concentrations under high relative humidity conditions (Jayaratne et al., 2018). The reason for this overestimation is that some aerosols can uptake water via hygroscopy. To solve this problem, several correction models have been proposed, such as a correction model based on the $\kappa$-Köhler theory (Crilley et al., 2018, 2020), multiple linear regression (Barkjohn et al., 2021; Nilson et al., 2022), and generalized additive model (Hua et al., 2021). Analysis of direct factors, such as temperature, relative humidity, and particle composition, can enhance the performance of low-cost sensors. In addition to these direct factors, we examine the impact of precursor gase $NO_2$, acting as a source of $PM_{2.5}$ emissions, on calibration performance in low-cost $PM_{2.5}$ sensors. In general, $PM_{2.5}$ arises by secondary formation from a chemical reaction between precursor gases, such as $NO_2$, in the atmosphere some distance downwind from the original emission source (Hodan et al., 2004). This study aims to identify the significance of the precursor $NO_2$ and evaluate its potential for improving the performance of low-cost $PM_{2.5}$ sensors. To this end, we considered two machine learning methods, Multiple Linear Regression (MLR) and Random Forest (RF), for calibration models using various feature vectors, including temperature, relative humidity, and $NO_2$. The trained MLR and RF models were evaluated on the test set, and their performance was compared. From an implementable perspective on $NO_2$ data, we investigated the feasibility of using data from distant $NO_2$ regulatory instruments due to the questionable data quality of low-cost $NO_2$ sensors. The results of our study showed that incorporating distant $NO_2$, in addition to temperature and relative humidity, into RF models yields lower errors than RF models that only include temperature and relative humidity.

## 2 Methods

### 2.1 Measurement data

#### 2.1.1 PurpleAir PA-II Units

The PurpleAir PA-II Outdoor air quality monitor was developed for measuring particulate matter of various sizes. PA-II units can measure various particulate matter as well as temperature, relative humidity, and barometric pressure. PurpleAir also devel-

oped a crowdsourcing platform to share publicly gathered PM measurements obtained from all PA units. From the PurpleAir website (https://www.purpleair.com/map), we can observe and download data reported by all installed PA units.

A PA-II unit includes two identical PMS 5003 sensors. The PMS 5003 sensors based on a light scattering principle measure concentrations of $PM_{1.0}$, $PM_{2.5}$, and $PM_{10}$ in real-time. By counting the number of particles per each diameter range that flows through a fan at a rate of 0.1L/min. Based on number of particles counted per diameter, each sensor estimates $PM_{1.0}$, $PM_{2.5}$, and $PM_{10}$ concentrations and then averages the concentrations every 80 s [1]. The PA-II unit sends the averaged concentrations obtained from two PMS sensors (A and B) to the PurpleAir server without storing the data in the unit itself. The PA-II unit does not calibrate the data, which implies it just collects the measured data.

The PurpleAir website provides the following information about all PA-II units via a JSON formated file: a name, a unique ID, a latitude, a longitude, and an installation date. Each PA-II unit has two unique IDs for each of its PMS sensors A and B.

### 2.1.2 Air quality measurement data from EPA

Outdoor air quality data collected from across the U.S. is publicly available through the U.S. Environmental Protection Agency (EPA) website (https://epa.gov/outdoor-air-quality-data). The EPA has a description file for monitors, which includes state code, county code, site number, location (latitude and longitude), parameter code, parameter occurrence code (POC), and last method. A combination of state code, county code, and site number can uniquely identify a monitoring site. For example, a monitoring station located at Bakersfield, CA has a state code of 06, a county code of 029, and a site number of 0014. The parameter code is an air quality system (AQS) code corresponding to the parameter measured by a monitor. For example, parameters regarding $PM_{2.5}$ and $NO_2$ are 88101 and 42602, respectively. A POC is used to identify an instrument among multiple ones with the same parameter code at a site. For example, two FRM instruments with a parameter of 88101 at the Bakersfield site are used to measure daily $PM_{2.5}$ concentrations and are identified with POC 1 and 2. The last method descriptor describes the measurement scheme used by the monitor for its most recent sample.

Monitoring ambient air quality for purposes of determining compliance with the U.S. National Ambient Air Quality Standards (NAAQSs) requires the use of either FRMs or FEMs. FRM and FEM instruments are accepted for methods for monitoring the NAAQS pollutants, such as particulate matters ($PM_{2.5}$ and $PM_{10}$), $NO_2$, $SO_2$, $O_3$, and CO. Hourly measurements of $PM_{2.5}$ and $PM_{10}$, as well as other pollutants such as $NO_2$, $SO_2$, $O_3$, and CO, obtained from FEM and non-FEM instruments can be downloaded via the EPA's application programming interface (https://aqs.epa.gov/data/api) (U.S. EPA, 2011). Daily measurements of $PM_{2.5}$ obtained from an FRM instrument are also available.

### 2.1.3 Selection of PA-II units and reference monitoring sites

To investigate the performance of a PA-II unit itself and evaluate its calibration, we focused on PA-II units that are installed close to an EPA monitoring site (i.e. reference site) that provides reliable hourly $PM_{2.5}$ concentrations. We use the location information of the PA-II units and reference monitors to find PA-II and reference monitor pairs that are located less than 100 m from each other(Wallace et al., 2021). Among the identified pairs, we selected a monitoring site, located at Rubidoux, CA, that

---

[1] After May 30, 2019, the averaging time is changed from 80 s to 120 s.

has 14 PA-II units as pairs and can measure other pollutants such as $NO_2$ on an hourly basis. The monitoring site is identified by a state code of 06, a county code of 065, and a site number of 8001 (i.e., 06-065-8001). This monitoring site is located in an urban residential area within the south coast air basin at an elevation of 248 m. Air pollutants from the Los Angeles and coastal areas are transported to this air basin, which is known to have poor ventilation and may experience air stagnation during the early evening and early morning periods. Local air pollution includes NOx from diesel trucks, since the city of Jurupa Valley, which includes the community of Rubidoux, is a main transportation corridor for diesel trucks serving three air cargo terminals and the ports of Los Angeles and Long Beach.

Table 1 describes information about the 14 PA-II units, such as their IDs, location (latitude and longitude), sensor name, start time of measurement, end time of measurement, and non-operating months[2]. While we present the ID for only PMS sensor A of each PA-II unit, the ID of PMS sensor B is the ID of PMS sensor A plus 1. The geographic information on 14 PA-II units and the monitoring site is shown in Figure S1. Distances between PA-II units and the monitoring site are shown in Table S1. The minimum and maximum distance between a PA-II unit and the monitoring site is less than 10 m and 100 m, respectively.

Based on the non-operating months of the PA-II units found, we selected an appropriate period of sample data from Jan. 2018 to Dec. 2019 (24 months). Among the 14 identified PA-II units, we chose several that had more than 23 months of valid measurement data during the period selected for study. The selected units are RIVR_Co-loc2, 3, 5, 6, 7, and 8, which we call PA-II 2, 3, 5, 6, 7, and 8, respectively.

Before using $PM_{2.5}$ data from the PA-II units, we checked the unit's data quality. We calculated the correlation among the selected PA-II units, considering both PMS 5003 sensors for each PA-II unit for the correlation analysis. Since these PA-II units are closely located, $PM_{2.5}$ data should be highly correlated. Figure 1 shows the correlation results for all PMS 5003 sensors included in the PA-II units. The numbers on each axis represent the number of the selected PA-II units. Boxes to the left and right of each number indicate PMS sensors A and B for its corresponding PA-II unit, respectively. The PMS sensor A of PA-II unit 2, PMS sensors A and B of PA-II unit 5, and PMS sensor A of PA-II unit 6 all have a poor correlation with other PMS sensors. In addition, sensor A of PA-II unit 3 has slightly poor correlation with other sensors. Based on these results, we selected PA-II units 7 and 8.

### 2.1.4 Data preprocessing of PA-II units

The PA-II units selected for study are long-term installations, i.e., they have been in operation for more than two years. Therefore, PA-II units may have abnormal data due to failure and aging drift, so data quality control is required before calibrating the PA-II units. The quality control (QC) measure has been shown to be important for developing correction models of PA-II units (Barkjohn et al., 2021). They performed a QC measure for obtaining daily $PM_{2.5}$ measurement data, but we applied the QC measure to obtain hourly $PM_{2.5}$ measurement data. The QC measure has the following 3 steps: i) data from both channels A and B was removed when either channel A or B had a missing value, ii) data with abnormal temperature or relative humidity values was removed, and iii) data from channels A and B were compared. In the first step, when we calculate 1-hour averages of $PM_{2.5}$ measurements generated with 2 min (or 80 sec) intervals, we remove the 1-hour average if the number of $PM_{2.5}$ mea-

---

[2]We define non-operating month as the month, when the number of days without the measurement data is larger than 10 days.

surements is less than 27 (or 40). We considered two different measurement intervals for a PA-II unit because its old interval had been 80 sec until May 30, 2019. Its current interval is 2 min. After calculating 1-hour average data, we removed all data points for the 1-hour interval, where either sensor A or B had a missing value. The second step deals with temperature and RH data. PA-II units occasionally report extremely high or low values of temperature and relative humidity that are inaccurate. Therefore, we removed the data points whose corresponding time interval contained unrealistic measurement of temperature or relative humidity. In this study, the acceptable ranges of temperature and RH are (0 °F, 200 °F) and (0%, 100%), respectively. Once the unacceptable data points were removed, we calculated the 1-hour average for temperature and RH. The last step was to compare results sensors A and B in a PA unit to check data consistency. To do this, we used symmetric percentage error (SPE) as follows:

$$SPE = \frac{2(|PM_{2.5}^A| - |PM_{2.5}^B|)}{|PM_{2.5}^A + PM_{2.5}^B|}, \tag{1}$$

where $PM_{2.5}^A$ and $PM_{2.5}^B$ are hourly averaged PM$_{2.5}$ concentrations from sensors A and B in the same PA-II unit, respectively. We removed the relevant data points with SPE larger than 0.61, which is 2 standard deviation. This value of SPE threshold has been used for 24-hr average PM$_{2.5}$ concentrations (Barkjohn et al. (2021)), but we use it here for 1-hour averaged PM$_{2.5}$ concentrations. The number of data points processed for each pre-processing step in PA-II 7 is summarized in Table S2.

## 2.2 Instrument intercomparisons

The monitoring site we considered has an FRM instrument and a BAM-1020 instrument with the parameter of 88502. These instruments produce daily and hourly PM$_{2.5}$ measurement data, respectively. Since we measure the PA-II units at intervals much shorter than a full day, it is much more reasonable to compare the PM$_{2.5}$ measurement of PA-II units with that of a BAM-1020 instrument with a shorter measurement interval, rather than that of an FRM instrument for evaluating the accurate calibration performance of PA-II units. However, we face the limitation that a BAM-1020 instrument can be classified as a non-FEM-compliant device. Therefore, our approach for analyzing PA-II units to appropriately resolve these issues is as follows: we compared the BAM-1020 instrument's readings with daily PM$_{2.5}$ concentrations collected from an FRM instrument to ensure the BAM-1020 provides an acceptable level of performance as an FRM instrument, which is enough to assess the calibration performance of PA-II units. According to this affirmative observation, the BAM-1020 instrument can be used to evaluate the calibration performance of low-cost PM$_{2.5}$ sensors by comparing its readings with hourly PM$_{2.5}$ measurement data of PA-II units.

We compared daily and hourly PM$_{2.5}$ measurement data obtained from an FRM and BAM-1020 intruments and PA-II 7 unit. Table 3 shows summary statistics of daily and hourly PM$_{2.5}$ measurement data from FRM and BAM instruments and PA-II 7 [3]. These data suggest that a BAM-1020 instrument using non-FEM methods compares well to the statistics achieved with the FRM method. However, the measurements are not enough to evaluate how similar the performance of the BAM-1020

---

[3]A PMS 5003 sensor that collects PM$_{2.5}$ concentrations from within a PA-II unit exhibits a maximum consistency error of $\pm 10 \, \mu g/m^3$ at 0-100 $\mu g/m^3$ and $\pm 10\%$ at 100-500 $\mu g/m^3$. The sensor reports PM$_{2.5}$ concentrations as integer values on a per-second basis. A PA-II unit generates readings of its own PM$_{2.5}$ concentrations by averaging its 1-second PM$_{2.5}$ concentrations over 80 (or 120) seconds.

is to that of the FRM instrument. Hence, this study compared the performance of two instruments using a linear fitting scheme. Figure 2 shows the calibration performance using linear regression. The $R^2$, slope, and intercept are 0.896, 0.923, and 0.741, respectively. Also, the value of RMSE is 2.211 μg/m$^3$. The BAM-1020 is close to an FEM instrument with the parameter
of 88101. In order for the BAM-1020 to attain the 88101 code in terms of performance, the following conditions must be satisfied: $R^2$ is larger than 0.9, a slope is larger than 0.9 and less than 1.1, and an absolute value of the intercept is less than 2.0. Slope and intercept are satisfied with the requirement, while $R^2$ does not meet the condition very slightly. Nonetheless, the BAM-1020 instrument provides an acceptable level of performance to evaluate the calibration performance of PA-II units on an hourly basis.

Compared to the FRM and BAM-1020 instruments, the PA-II 7 unit overestimates the maximum daily PM$_{2.5}$ concentrations. Additionally, the mean daily PM$_{2.5}$ concentration from the PA-II 7 unit was higher than that of the FRM and BAM-1020 instruments. These results show that the PA-II unit has a good correlation ($r$) with the FRM instrument for the two-year period of interest, since its value is very close to 1. However, a comparison of metrics from the FRM instrument and the PA-II 7 unit did not correlate as favorably.

Next, we compared the PA-II unit's hourly PM$_{2.5}$ data with that of the BAM-1020 instrument over the course of the same two-year period. We did not consider the FRM instrument for exploring hourly PM$_{2.5}$ measurement data, since it only produces daily concentrations. The PA-II unit's maximum hourly PM$_{2.5}$ measurement was almost twice that of the BAM-1020. In other words, the PA-II unit overestimates hourly PM$_{2.5}$ concentrations. Figure 3 shows the comparison of PM$_{2.5}$ measurement data obtained from the BAM-1020 and the selected PA-II 7 unit, as well as temperature and relative humidity measured from the
selected PA-II 7 unit during winter season (from Dec. 2018 to Feb. 2019). The PA-II 7 unit showed a similar trend of PM$_{2.5}$ concentration measurements to that of the BAM-1020 instrument, but it generally overestimated hourly PM$_{2.5}$ concentrations more often than the BAM-1020.

In addition, we compared the hourly PM$_{2.5}$ concentrations of the PA-II unit with that of the BAM-1020 instrument in terms of RMSE, MSE, MAE, and $r$. The results are as follows: RMSE of 6.194 μg/m$^3$, MSE of 38.369 μg/m$^3$, MAE of 7.919
210  μg/m$^3$, and $r$ of 0.876. The PA-II unit had a good correlation with the BAM-1020 instrument based on $r$. However, other metrics, such as RMSE, MSE, and MAE, did not correlate well.

### 2.3   Feature selection for calibration models

Temperature and relative humidity have been identified in previous studies as key factors for effective calibration. In particular, relative humidity has been shown to affect low-cost PM sensors under high relative humidity conditions. Furthermore, few
papers have considered NO$_2$ in calibration models (Hua et al., 2021) because NO$_2$, which is known to be a precursor to the formation of PM$_{2.5}$ through chemical reactions in the atmosphere, may indirectly affect PM$_{2.5}$ concentrations. Therefore, we investigated the suitability of temperature, relative humidity, and NO$_2$ for the calibration of the PA-II 7 unit.

To identify the independent variables relevant for calibration, we conducted a correlation analysis involving PM$_{2.5}$ measurements from BAM-1020 and PA-II 7 unit readings, as well as temperature and relative humidity data, spanning a two-year
period. The results are illustrated in Figure S2. The highest correlation was observed between PM$_{2.5}$ from BAM-1020 and

PA-II 7 unit, followed by $NO_2$ measurements. Subsequently, relative humidity and temperature exhibited the next level of correlation. As a result, we have identified temperature, relative humidity, and $NO_2$ as the selected candidate features.

To explore the potential for enhancing the calibration performance of low-cost PM sensors using temperature, relative humidity, and $NO_2$ as features, we conducted linear fitting. Before considering temperature, relative humidity, and $NO_2$, we
evaluate the monthly performance based on hourly $PM_{2.5}$ data from the PA-II 7 unit compared to the BAM-1020 instrument. Table 2 shows the value of $R^2$, RMSE, and MAE of hourly $PM_{2.5}$ measurement data from the PA-II 7 unit compared to that of the BAM-1020 instrument and the corresponding slope and intercept of each optimal linear fitting. During the months of Nov., Dec., and Jan., the PA-II unit is shown to have a high correlation, $R^2$ of 0.813 to 0.936, with the BAM-1020 instrument. This result is supported by the field evaluation of PA-II units taken by the Air Quality Sensor Performance Evaluation Center
(AQ-SPEC) during the period of Dec. 2016 - Jan. 2017, which showed the value of $R^2$ as being 0.868 to 0.921 when the PA-II units were compared with the FEM. (Sayahi et al., 2019) showed that PMS sensors have a high correlation with tapered element oscillating microbalances (TEOM) instruments during the winter season by providing $R^2$ of 0.866 to 0.892. That is, the hourly $PM_{2.5}$ measurement data from PA-II units seem to be highly correlated with that of FEM instruments during the months of November, December, and January, which implies the $PM_{2.5}$ measurement performance of PA-II is reliable, especially during
winter seasons. These months have different slopes and intercepts; for example, Jan. 2018 has a slope of 0.502 and an intercept of 3.898, while Jan. 2019 has 0.397 and 1.961, respectively.

On the other hand, the PA-II 7 unit has a correlation lower than 0.6 for months of Jul. and Sep. 2018 as well as Aug., Sep., and Oct. 2019. These months, except Sep. 2019, have larger RMSE values compared to other months over the two-year period, which need to be calibrated.

For multiple features, such as temperature, relative humidity, and $NO_2$, we used an MLR approach for regression analysis of PA-II units compared to the BAM-1020 instrument. A per-month analysis was conducted based on hourly $PM_{2.5}$ measurements from the PA-II 7 unit under several feature vectors, such as ($PM_{2.5}$), ($PM_{2.5}$, T), ($PM_{2.5}$, RH), ($PM_{2.5}$, $NO_2$), ($PM_{2.5}$, T, RH), and ($PM_{2.5}$, T, $NO_2$), where T and RH represent temperature and relative humidity, respectively. For notational simplicity, we defined the above feature vectors ($PM_{2.5}$), ($PM_{2.5}$, T), ($PM_{2.5}$, RH), ($PM_{2.5}$, $NO_2$), ($PM_{2.5}$, T, RH), and ($PM_{2.5}$, T, $NO_2$) as 1,
2, 3, 4, 5, and 6, respectively. Figure 4 shows the $R^2$ and RMSE results of multiple linear regression for selected months with the above varying feature vectors. We considered feature vector 1 as a baseline for comparison among other feature vectors. On Jan. 2018, feature vector 5, referring to temperature and relative humidity, had little effect on the regression performance of $R^2$ and RMSE. The amount of $R^2$ increase by feature vector 5 from the baseline was around 0.001, and the amount of RMSE decrease was $0.038\,\mu g/m^3$. In the case of feature vector 6, including $NO_2$ instead of RH, $R^2$ increased from the baseline by
0.015, while RMSE was improved by $0.518\,\mu g/m^3$. Similarly, for Apr. 2018, $R^2$ (or RMSE) for feature vector 5 increased (or decreased) by 0.01 (or $0.072\,\mu g/m^3$) compared to its baseline. $R^2$ and RMSE for feature vector 6 increase by 0.05 and decrease by $0.52\,\mu g/m^3$ from the baseline, respectively. For regressions in Aug. and Sep. 2019, an increase in $R^2$ was larger than 0.17 when feature vector 6 was considered, but it was less than 0.07 when feature vector 5 was considered. These remarkable results suggest that $NO_2$ is generally a key factor that can improve the performance of PA-II units over a year, even though

the enhancement by $NO_2$ does not meet the values of 0.7 of $R^2$ and 3.5 μg/m$^3$ of RMSE during certain months, such as July 2018, August 2019, October 2019.

## 2.4 Calibration methods

A per-month analysis with a combination of features, including T, RH, and $NO_2$, showed an effect on calibration for the PA-II unit. However, it is challenging to use the per-month linear fitting result to calibrate PA-II units because each month has a different slope and intercept defined for the linear fitting. Moreover, their values exhibit a change over the years. For example, notably, the linear fitting result in Apr. 2018 exhibited a higher RMSE than the fitting result in Apr. 2019. On the contrary, the calibration performance in Aug. 2018 was worse than that in Aug. 2019.

We used a machine learning approach to develop a calibration model, employing two machine learning algorithms, such as multiple linear regression (MLR) and random forest (RF). For both calibration methods, we considered various combinations of features, including $PM_{2.5}$ measured from a PA-II unit, temperature, relative humidity, $NO_2$, and their multiplicative interaction terms.

### 2.4.1 Multiple linear regression (MLR)

An MLR method can be expressed as follows:

$$\hat{y} = \beta_0 + \beta_1 x_1 + \cdots + \beta_n x_n, \tag{2}$$

where $\hat{y}$ represents a response, $n$ is the number of predictor variables, $\beta_i$ for $i = 0, 1, \ldots, n$ are regression coefficients, and $x_i$ for $i = 1, 2, \ldots, n$ represent predictor variables (called features). Using a linear equation with multiple variables, we investigated the relationship between features and a response.

All features in an MLR method should be independent. However, many studies have considered $PM_{2.5}$, temperature, and RH, which are not independent (Magi et al. (2019); Mailings et al. (2020)). Some studies have introduced multiplicative interaction terms (i.e., $PM_{2.5} \times RH$) to exploit interdependence between features (Barkjohn et al. (2021)). We also consider multiplicative interaction terms in this study.

We use $PM_{2.5}$ concentrations obtained from a reference monitor as the response. As predictor variables, we consider multiple features, such as $PM_{2.5}$ measurement data from a PA-II unit, temperature, relative humidity, $NO_2$, and their multiplicative interaction terms (i.e., $PM_{2.5} \times RH$, $T \times RH$, $PM_{2.5} \times RH \times T$).

### 2.4.2 Random forest (RF)

An RF is an ensemble of $K$ regression trees. Each regression tree is trained with a bootstrap sample of an original training dataset. The output of an RF is the aggregation of regression trees, i.e., averaging estimates over all trees. Each regression tree is grown by selecting random $m$ features among $M$ input features at each possible split. The best cut is calculated at the randomly chosen features. Optimal cuts can be achieved using the Classification and Regression Trees split criterion (CART),

which compares the variance of the uncut node and one of all possible cuts along $m$ directions. Every tree is fully grown with these splits (Breiman, 2001).

## 2.5 Performance evaluation metrics

In this study, we examined the root mean square error (RMSE), mean squared error (MSE), mean absolute error (MAE), and Pearson correlation coefficient $r$ between daily $PM_{2.5}$ data from the FRM instrument and that from the PA-II units. In the cases
of the RMSE, MSE, and MAE, the lower its value is, the better the performance or the lower the difference in measurement data between the FRM instrument and the PA-II units. The Pearson correlation coefficient is a metric measuring a linear correlation between two variables. It is a number between -1 and 1 that measures the strength and direction of their relationship. As the coefficient approaches an absolute value of 1, the values of measurement data from the FRM instrument and the PA-II units become more similar. These performance metrics are expressed as follows:

$$RMSE = \sqrt{\frac{1}{n}\sum_{i=1}^{n}(x_i - y_i)^2}, \tag{3}$$

$$MSE = \frac{1}{n}\sum_{i=1}^{n}(x_i - y_i)^2, \tag{4}$$

$$MAE = \frac{1}{n}\sum_{i=1}^{n}|x_i - y_i|, \tag{5}$$

where $x_i$ represents 1-hour averaged (24-hour period) sensor $PM_{2.5}$ concentrations for the $i$th hour (day) ($\mu g/m^3$), $y_i$ represents 1-hour averaged (24-hour period) FRM or BAM-1020 $PM_{2.5}$ concentrations for the $i$th hour (day) ($\mu g/m^3$), and $n$ is the number
of data points.

## 3 Results and discussions

### 3.1 Calibration performance

The two-year dataset was divided into training and test sets at a 1:1 ratio, meaning the measurement data in the years 2018 and 2019 were used for training and testing, respectively. We used the training set to learn calibration models based on MLR
and RF, and then used the test set to evaluate the calibration performance in terms of RMSE, MAE, and $R^2$. A calibration performance for the PA-II 7 unit using MLR and RF methods was compared with several features, including temperature, relative humidity, and $NO_2$, as well as their multiplicative terms.

#### 3.1.1 MLR-based calibration model

Recently, calibration methods have employed multiplicative interaction terms, such as $PM_{2.5} \times RH$ and $T \times RH$. In our MLR
models, we considered both additive and multiplicative interaction terms. The additive terms in our models include raw Purple-Air $PM_{2.5}$, T, RH, and $NO_2$. We considered multiplicative interaction terms that involve less than four additive terms when

NO$_2$ was not included (i.e., we consider PM$_{2.5}\times$T$\times$RH), and less than three additive terms when NO$_2$ is included. There are 95 combinations of features. Out of 95 combinations tested, only 52 combinations had a p-value of less than 0.05. Of those, we select 21 combinations, among 52 combinations, by increasing the number of additive terms and the number of multiplicative interaction terms and identifying the combinations with the lowest RMSE among the same numbers of additive terms and multiplicative interaction terms. The selected combinations were shown in Table 4.

The calibration results of the PA-II 7 unit for test datasets using the MLR method with 21 selected combinations are presented in Table 5. Multicollinearity is a known issue with MLR models, as it can cause instability. One common method to diagnose this issue is to use the variance inflation factor (VIF) test for multicollinearity (Mansfield and Helms , 1982). Out of the 21 combinations tested, most VIF values were less than 5, indicating the absence of collinearity issues.

When a single additive term, such as T or RH, was applied, the RMSE values for two combinations, #2 and #3, improved by more than 0.208 $\mu g/m^3$, compared to considering only PM$_{2.5}$. The inclusion of an additive RH term in an MLR yielded a lower error than an additive T term did, since both RMSE and MAE for combination #3 were less than those for combination #2. The MLR model with PM$_{2.5}$, the single additive term with RH, and its multiplicative interaction term with PM$_{2.5}$ yielded similar RMSE and MAE to the MLR model using PM$_{2.5}$ and two meteorological variables, such as T and RH, as demonstrated by the results of combinations #4 and #5. When we considered two meteorological variables and incorporated four multiplicative interaction terms, such as PM$_{2.5}\times$T, PM$_{2.5}\times$RH, and T$\times$RH, the MLR model resulted in the lowest error with an RMSE of 4.151 μg/m$^3$ and an MAE of 3.023 μg/m$^3$, compared to all combinations generated from PM$_{2.5}$, T, RH, and their multiplicative terms.

The MLR model of combination #10 with PM$_{2.5}$ and NO$_2$ had an RMSE of 4.424 μg/m$^3$, which was lower than that of the MLR model with only PM$_{2.5}$, whose RMSE was 4.513 μg/m$^3$, but larger than that of combination #2 with a single environmental variable and an RMSE of 4.305 μg/m$^3$. This implies that the addition of a single multiplicative term in that model has no performance enhancement. However, when the additive term T is incorporated into an MLR model with PM$_{2.5}$ and NO$_2$, an RMSE of 3.997 μg/m$^3$ can be achieved, which is lower than the values of all combination cases, not including NO$_2$, i.e., combinations #1 to #9. Coefficients of PM$_{2.5}$, T, and NO$_2$ in the MLR model, including T and NO$_2$, were around 0.446, 0.110, and 0.112, respectively. The temperature had more impact on error than relative humidity when considering NO$_2$. Considering both temperature and relative humidity together with NO$_2$ may cause a non-zero correlation of relative humidity with other factors due to a p-value of 0.083. When some multiplicative terms were additionally integrated into T, RH, and NO$_2$, the MLR calibration models passed a p-value test. The model based on combination #18 with four additive terms, i.e., PM$_{2.5}$, T, RH, and NO$_2$, and multiplicative interaction terms, including PM$_{2.5}\times$RH and T$\times$RH, achieved the lowest RMSE of 3.912 μg/m$^3$. Considering multiplicative terms with T and RH had little effect on calibration performance as shown in the results of combinations #15, #19, and #20. From these results, we conclude that considering NO$_2$ together with meteorological variables and their multiplicative terms or a single variable, such as temperature, can improve the calibration performance of PA-II units.

### 3.1.2 RF-based calibration model

This study validated performance of RF-based calibration for PA-II units with 95 combinations of predictor variables mentioned in the previous subsection. An RF was implemented using the scikit-learn package in Python. An RF has several hyperparameters, such as n_estimators, max_depth, min_samples_leaf, and max_features, that need to be set for the best performance over each combination of features. For this study, the hyperparameters were tuned with a random search method by 5-fold cross-validation based on the training set. For a random search, the number of trees (n_estimators) was set to 10, 20, 50, 100,

200, and 400. The range of max_depth was set to 2, 4, 6, 8, 10, 16, and None. The range of min_samples_leaf was set to 1, 2, 3, 4, and 5. The range of min_samples_split was set to 2, 3, 5, 7, and 10. The range of max_features was set to None.

We selected 22 combinations according to the above mentioned method. The selected combinations were listed in Table 6. Table 7 summarizes calibration results, including $R^2$, RMSE, and MAE of test sets for PA-II units using the RF method with the selected combinations of features.

Like the MLR method, the RF method showed better performance on the training set than on the test set. Some combinations had larger RMSE differences than 0.6 $\mu g/m^3$ between training and test sets, while others have differences smaller than 0.4 $\mu g/m^3$. We note that some combinations with multiplicative terms showed significant RMSE differences between two datasets, which might have occurred because of overfitting the training dataset. Nonetheless, the RF models with the other combinations had lower RMSE than the model using only $PM_{2.5}$. Considering a single environmental variable together with $PM_{2.5}$ improved

the calibration performance in terms of values of RMSE and MAE compared to the RF model with only $PM_{2.5}$. Specifically, RH had more significant impact on the performance enhancement of the RF calibration model than T as seen in the results of combinations #2 and #3. Including the additional multiplicative term of $PM_{2.5} \times RH$ had an insignificant effect on RMSE compared to the RF model with $PM_{2.5}$ and RH. Both meteorological variables together, i.e., combination #5, yielded lower RMSE in the training set compared to the RF model with $PM_{2.5}$ and RH, i.e., combination #3, but similar RMSE in test set. In

contrast to MLR models, more than one multiplicative term, i.e., combinations #6 to #9, bring about insignificant difference in RMSE compared to considering a single meteorological variable. When we analyze calibration methods without $NO_2$, the RF model with $PM_{2.5}$, T, and RH improved RMSE by 0.117 $\mu g/m^3$, compared to the best MLR model.

Utilizing $NO_2$ on RF models had different effects on calibration performance, depending on the combinations of predictor variables. The RF model of combination #10 with the additional $NO_2$ term resulted in an RMSE of 4.434 $\mu g/m^3$, which was

370 little improvement compared to combination #1 with only $PM_{2.5}$ and an RMSE of 4.439 $\mu g/m^3$. The RF model with $PM_{2.5}$ and $NO_2$ had a larger RMSE than the MLR model with the same features, but the difference was not significant, it did not show enough performance improvement to warrant adding the multiplicative term of $PM_{2.5} \times NO_2$ from combination #10. Adding single or two meteorological variables to RF models of combinations #12 and #16 lead to remarkable performance enhancement over combination #10, with RH, RMSE decreasing by 0.462 $\mu g/m^3$. Furthermore, RMSE dropped an additional 0.130 $\mu g/m^3$

when T was added as an additional feature. The combinations consisting of one or more multiplicative interaction terms resulted in either an insignificant improvement or a slight decline in the performance in terms of RMSE and MAE when compared with

combination #16 consisting of $PM_{2.5}$, T, RH, and $NO_2$. In other words, there is no need to consider multiplicative interaction terms when we use the RF model, because there is no outstanding performance improvement.

As with the MLR method, it was shown that including $NO_2$ as a consideration in RF methods can improve calibration performance. Moreover, by integrating two additional variables, such as T and RH, even better calibration performance can be achieved.

The RF method was shown to have a better performance than the MLR method when $NO_2$ was not considered. From the viewpoint of RMSE, the best performance from MLR and RF methods was 4.151 µg/m$^3$ and 4.014 µg/m$^3$, respectively. However, when we consider $NO_2$, the best MLR model is not significantly different from the best RF model. For instance, the RMSE values from the best MLR and RF models were 3.912 µg/m$^3$ and 3.840 µg/m$^3$, respectively. Their corresponding $R^2$ values differ slightly, since their gap is only 0.008. Nonetheless, the MAE of 2.777 µg/m$^3$ achieved from the best MLR is lower than that achieved by the best RF, which is 2.831 µg/m$^3$. From these results, we conclude that better calibration can be obtained by considering $NO_2$ additionally. Furthermore, when $NO_2$ is considered, the MLR model can enhance calibration performance without the need for an RF model.

## 3.2 Effect of distant $NO_2$ on calibration performance

In the previous subsections, it was demonstrated that including $NO_2$ as a consideration can effectively improve the calibration performance of PA-II units. However, it is not always feasible to have an $NO_2$ instrument with high accuracy collocated with a low-cost PM sensor. Instead, an alternative approach is to collocate a low-cost $NO_2$ sensor with a PA-II unit, but this approach is hindered by the unreliability of $NO_2$ sensors. To address this issue, we investigated the usefulness of using data from distant $NO_2$ instruments installed with PA-II units for the calibration algorithm.

We selected two monitoring sites that measure $NO_2$ near the Rubidoux site. Two monitoring sites identified were 06-065-8005 and 06-071-0027. The distances between the two monitoring sites and the Rubidoux site are 7.05 km and 18.87 km, respectively. The correlations of $NO_2$ measurements obtained from the Rubidoux site with that of 06-065-8005 and 06-071-0027 were 0.895 and 0.621, respectively. The site 06-065-8005 had $NO_2$ measurements that are much more highly correlated with the Rubidoux site compared with those from the site 06-071-0027. This result can occur when the distance from the Rubidoux site to the site 06-065-8005 is shorter than it is to the site 06-071-0027.

To evaluate the usefulness of distant $NO_2$ measurements on the calibration of a low-cost PM sensor, we used $NO_2$ data measured from monitoring sites near the PA-II 7 unit as a test dataset, rather than data from the collocated Rubidoux site. When we trained calibration models with the measurements from the PA-II 7 unit over 2018, we used highly accurate $NO_2$ concentrations measured by FEM instruments at the Rubidoux site. Subsequently, to verify the trained calibration models, we utilized a separate test dataset featuring distant $NO_2$ measurements taken by FEM instruments at sites 06-065-8005 and 06-071-0027. We considered this scenario to evaluate our proposed calibration models, previously trained with collocated $NO_2$ concentrations and distant $NO_2$ concentrations, when collocated $NO_2$ measurements cannot be collected.

Table 8 shows calibration performance using MLR and RF methods with $NO_2$ collected from the air quality monitoring sites near the PA-II unit. In the case of MLR methods used with 06-065-8005 data, the difference in RMSE between $NO_2$ data

obtained from a collocated $NO_2$ instrument, called collocated $NO_2$, and a distant $NO_2$ instrument, called distant $NO_2$, was less than $0.06\,\mu g/m^3$ for every selected combination defined in previous two subsections for the MLR and RF methods. All MLR models using distant $NO_2$, except combinations #10 and #11, yielded lower errors than all MLR models without $NO_2$ as shown in Table 5. For example, the worst RMSE of the MLR methods using distant $NO_2$ data (except combinations #10 and #11) was 4.018 $\mu g/m^3$, while the best RMSE without $NO_2$ was 4.151 $\mu g/m^3$. Like RMSE, other metrics, such as $R^2$ and MAE, also showed a calibration performance enhancement for these combinations with distant $NO_2$.

When we used an MLR algorithm with $NO_2$ data, the result of the calibration performance for the monitoring site 06-071-0027 showed a new aspect from that of 06-065-8005. All MLR methods using distant $NO_2$ data from site 06-071-0027 had a higher RMSE than the MLR algorithm was based on data that did not include $NO_2$ data from the collocated Rubidoux instrument, which had an RMSE of 4.513 $\mu g/m^3$ as shown in Table 5. This result can be explained by comparing correlation of $NO_2$ measured from the Rubidoux site with measurements from site 06-065-8005 as well as site 06-071-0027. The $NO_2$ correlation between Rubidoux measurements and site 06-065-8005 was 0.895, while the correlation with site 06-071-0027 was 0.621. These results shows that 06-065-8005 data is much more correlated with the Rubidoux site in terms of $NO_2$.

In the case of RF models, the use of the distant $NO_2$ data from site 06-065-8005 increased RMSE compared to using collocated $NO_2$ data, but not significantly, since the maximum gap of RMSE values for all feature vectors considered was just 0.060 $\mu g/m^3$. Similar to the MLR method, all RF models referring to distant $NO_2$ from site 06-065-8005, except combinations #11, resulted in a better calibration performance than was seen in combination #1 without $NO_2$ which had an RMSE of 4.439 $\mu g/m^3$ shown in Table 7. Other metrics, such as $R^2$ and MAE, also showed a calibration performance improvement. In the case of RF models using data from site 06-071-0027, calibration performance for each combination was degraded compared to the corresponding combination using collocated $NO_2$, which had similar results of the MLR model. As we explained previously, the higher the correlation of $NO_2$ measurements from the Rubidoux site with measurements from sites 06-065-8005 and 06-071-0027, the better the calibration performance of the RF model; that is, all combinations with distant $NO_2$ from 06-065-8005 provide a lower RMSE than those from 06-071-0027. Moreover, when we consider 06-065-8005 having a high correlation of $NO_2$ with the expensive $NO_2$ instrument collocated with the PA-II 7 unit, the best RMSE for all combinations using the RF model is slightly lower than that based on the MLR method.

In the case of 06-065-8005, RF models using distant $NO_2$ resulted in lower, but insignificant, RMSE values, compared to MLR models using distant $NO_2$. From these results, we draw the conclusion that the use of $NO_2$ collected from distant instruments with a high correlation with a collocated $NO_2$ site of PA-II units can improve the PA-II unit's calibration performance. Furthermore, both MLR and RF models can be good calibration models when distant $NO_2$ is considered. This is different from the conclusion that calibration performance of RF models is better than MLR models (Zimmerman et al., 2018).

## 3.3 Applicability of other PA-II units

We evaluated PA-II 8's calibration performance under the following three cases:

Case 1: Calibration model is learned with the measurements collected from the PA-II 8 in 2018 and calibration performance for the trained model is evaluated using data measured from the PA-II 8 in 2019.

Case 2: This is similar to Case 1, except that the calibration model is trained with the data measured from the PA-II 7 in 2018.

Case 3: The measurement data from the PA-II 8 with collocated $NO_2$ concentration in 2018 is used as a training dataset, while the data collected from the PA-II 8 with either collocated $NO_2$ or distant $NO_2$ concentration in 2019 is used as a test dataset.

In Case 1, we evaluated the calibration model's performance with a test dataset consisting of measurement data from the PA-II 8 in 2019. The calibration model is trained with data collected from the same PA-II 8 in 2018. Table 9 shows the calibration results of the PA-II 8 using an MLR method under two different cases: with and without $NO_2$. We selected the same feature vectors as defined in Table 4. We observed that $NO_2$ can enhance calibration performance because all MLR models using $NO_2$, except combinations #10 and #11, yield lower errors and larger $R^2$ than those without $NO_2$. This observation aligns with the results shown in Table 5. Additionally, compared to the calibration performance for PA-II 7 shown in Table 5, PA-II 8 shows slightly larger RMSE and MAE, but similar $R^2$.

In Case 2, we evaluated the calibration model's performance using a training dataset collected from PA-II 7 in 2018, and a test dataset collected from PA-II 8 in 2019. Table 10 shows calibration results for PA-II 8 using the MLR method under two different conditions, such as with and without $NO_2$. As with the observation in Table 9, $NO_2$ is the key factor enhancing calibration performance. With the exceptions of #10 and #11, all MLR models using $NO_2$ yield lower errors and larger $R^2$ than those without $NO_2$. It is important to compare this result with that shown in Table 5, as we used different test datasets. It could be expected that the much worse performance for all feature combinations listed in Table 10 is achieved than for every corresponding feature vector in Table 5, since the calibration model considered in Table 10 is tested with the data measured from the PA-II 8, whereas it is trained with the measurement data collected from the PA-II 7. $R^2$ values of all feature vectors in Table 10 are similar to those for each corresponding feature vector in Table 5. Unlike $R^2$, we observe larger RMSE and MAE when we populate the training dataset with measurements from PA-II 8 rather than PA-II 7. The maximum differences of RMSE and MAE for each feature vector in Tables 10 and 5 are 0.177 $\mu g/m^3$ and 0.196 $\mu g/m^3$, respectively.

The results shown in Table 9 and Table 10 support our conclusion that reliable and consistent PA-II units, which contain two PMS 5003 sensors with high correlation, demonstrate similar calibration performance. This implies that a proposed calibration method can be applied to reliable and consistent PA-II units generally.

Lastly, in Case 3, we evaluated the effect of collocated and distant $NO_2$ on PA-II 8 unit's calibration performance. Table 11 shows the results of MLR-based calibration model for the PA-II 8 when it is verified with the test data considering either collocated or distant $NO_2$. As we explained in Section 3.2, we considered two monitoring sites measuring $NO_2$ near the Rubidoux site. One site (ID 06-065-8005) had $NO_2$ measurements that are much more highly correlated with the Rubidoux site than those from the other site (ID 06-071-00247). We refer to the $NO_2$ concentrations measured from these two sites as "distant $NO_2$". Three columns, describing the values of $R^2$, RMSE, and MAE, of collocated $NO_2$ in Table 11 are exactly the

same as those of NO$_2$ included (i.e., collocated NO$_2$) in Table 9. In the case of site 06-065-8005 with high correlation with the Rubidoux site, the consideration of the distant NO$_2$ facilitates improvement of the calibration performance, since all MLR-based calibration models using distant NO$_2$, except combinations #10 and 11, produce lower errors and larger $R^2$ than those without NO$_2$. This result is similar to when we consider the collocated NO$_2$. However, we observe that adding distant NO$_2$ to the test dataset, which is not highly correlated to the NO$_2$ measurement from the reference site, deteriorates the calibration performance. This is likely because all combinations from #10 to #21 yield lower $R^2$ and greater errors than all combinations excluding NO$_2$, as shown in Table 9. This result is the same as the observation of the PA-II 7 unit's calibration results in Table 8.

Hence, these results we draw from Table 11 support the same conclusions we drew from Tables 9 and 10. Reliable and consistent PA-II units achieve similar calibration performance, and our proposed calibration model can be applied to these units generally.

### 3.4 Effect of training period

We evaluated the effect of the training period on calibration performances. We consider four different training periods (i.e., 3, 6, 9, and 12 months), and each training set is constructed as follows: The training sets all end at the close of 2018. Their start points are set in reverse order based on training periods. For example, for 3 months, the training set is from Oct. 2018 to Dec. 2018. Table S4 shows PA-II 7's calibration results using the MLR method for all four training periods. The 3-month training period has the worst performance. The 6- and 9- month training periods generated better performances than the 12-month training period. From a viewpoint of using NO$_2$, NO$_2$ can improve calibration performance in all four cases, compared to using only temperature and relative humidity. As the length of the training period increases, calibration performance improves.

### 3.5 Uncertainty analysis

We performed an uncertainty analysis of the MLR-based calibration model by using a bootstrapping technique on a test dataset. Table 12 shows statistics of uncertainty analysis for each feature vector and t-values between two feature vectors whose difference is the existence of NO$_2$. We selected 8 feature vectors with various independent variables to verify whether the addition of NO$_2$ affects the performance of our calibration model. The 4 feature vectors we considered are PM$_{2.5}$, PM$_{2.5}$, T, PM$_{2.5}$, RH, and PM$_{2.5}$, T, RH. We also added NO$_2$ to create four other feature vectors, PM$_{2.5}$, NO$_2$, PM$_{2.5}$, T, NO$_2$, PM$_{2.5}$, RH, NO$_2$, and PM$_{2.5}$, T, RH, NO$_2$. We generated 1,000 test sets using a bootstrapping technique with replacement. We evaluated mean and standard deviation values of RSME calculated over 1,000 test sets for each feature vector. In addition, we applied a t-test to verify the effectiveness of adding NO$_2$ to each feature vector. Consideration of NO$_2$ additionally reduces mean values of RMSE for all 4 feature vectors. Contrary to mean value, standard deviation of RMSE for every feature vector increases slightly with the addition of NO$_2$. We evaluated t-value for the mean values of RMSE for two feature vectors, with and without NO$_2$; for example, the t-value between PM$_{2.5}$ and PM$_{2.5}$, NO$_2$. Hence, we can evaluate 4 t-values. The Degree of Freedom (DoF) is 1,998, so the relevant p-values are much less than 0.00001. Therefore, the difference in the mean RMSE values of

the PM$_{2.5}$–included and PM$_{2.5}$-excluded groups is significant. From these results, we can conclude that the performance of the MLR-based calibration model can be enhanced with consideration of PM$_{2.5}$ concentrations.

## 4 Conclusions

The factors, directly affecting the performance of a low-cost PM sensor, including temperature, relative humidity, and particle composition, have been scrutinized for their impact on sensors' performance enhancement. Additionally, this study investigated the potential of NO$_2$, a precursor gas that gives rise to PM$_{2.5}$ through atmospheric chemical reactions, to improve performance of the calibration model. To this end, we used the PurpleAir PA-II unit, which contains two Plantower PMS 5003 sensors, as a low-cost PM$_{2.5}$ sensor. The PA-II units need to be typically installed close to reference monitoring sites measuring PM$_{2.5}$ concentrations and other pollutants, such as NO$_2$, in order to analyze their calibration. We identified a EPA-certified monitoring instrument whose deployed location is within close proximity to the installed location of 14 PA-II units, which satisfied the condition for co-location with a reference monitoring site. The monitoring site is located in Rubidoux, CA, USA. A study period of two years, i.e., from Jan. 2018 to Dec. 2019, was selected to include all seasons. Two units among 14 PA-II units were selected based on the availability of 23 months or more of measurement data from each PA-II unit, as well as its low intra-model variability through correlation analysis.

One of the two selected PA-II units was compared to FRM and BAM-1020 instruments based on daily and hourly PM$_{2.5}$ measurements. A comparison of the BAM-1020 instrument with the FRM instrument was also conducted on a daily PM$_{2.5}$ measurement basis to evaluate the performance of the BAM-1020. The BAM-1020 instrument had a slope of 0.923, an intercept of 0.741, and a $R^2$ of 0.896 against the FRM instrument, which implies it provides acceptable performance as a reference monitor for the calibration of low-cost PM$_{2.5}$ sensors. For a PA-II unit, the Pearson correlation coefficient against the BAM-1020 instrument was shown to be 0.928 on an hourly basis. The per-month analysis was conducted on hourly PM$_{2.5}$ measurements of the PA-II unit against the BAM-1020. Results showed the PA-II unit has a good correlation during the winter season, i.e., Nov., Dec., and Jan., with an $R^2$ value between 0.819 and 0.906, but a lower correlation during other months. The performance of the PA-II units was not notably affected by temperature or relative humidity (RH) during the winter months. Temperature and/or RH were found to improve $R^2$ during June and July 2018, but this effect in 2019 was not the same as in 2018.

A per-month analysis showed that NO$_2$ is a key factor that increased the value of $R^2$ during Sep. 2018, and Aug. and Sep. 2019. The effect of the addition of NO$_2$ for calibration of PA-II units was much larger when RH and temperature were considered together. In particular, NO$_2$ was shown to have more effect during months when the performance of PA-II units is moderate. It is expected that NO$_2$ can be used to improve the performance of low-cost PM$_{2.5}$ sensors, but the effect of NO$_2$ should be further investigated for various ambient conditions.

Two methods for calibrating PA-II units, the Multiple Linear Regression (MLR) and Random Forest (RF), were evaluated on a test set of one year of data. We considered additive and multiplicative terms in two calibration methods. The RF method yielded better performance than the MLR method because it provides a larger $R^2$ as well as smaller RMSE, and MAE when NO$_2$, called collocated NO$_2$, measured from the collocated monitoring site was not used for calibration. However, when

collocated $NO_2$ is considered, MLR models showed similar performance to RF models. When several features, such as $PM_{2.5}$, temperature, RH, $NO_2$, and their multiplicative terms, are considered together to calibrate $PM_{2.5}$ measurement data using the MLR method, the calibration performance was shown to increase remarkably compared to cases where only $PM_{2.5}$ are considered. For instance, the RMSE value decreased from 4.513 µg/m$^3$ to 3.912 µg/m$^3$. In RF models with collocated $NO_2$, inclusion of temperature and RH improved $R^2$, RMSE, and MAE by an increase of 0.018, a decrease of 0.172 µg/m$^3$, and 0.119 µg/m$^3$, respectively, compared to the best RF models without $NO_2$. Contrary to the MLR model, multiplicative interaction terms do not affect calibration performance with a certain direction, compared to those without $NO_2$; some combinations of features provide slight enhancement, while the others cause worse performance.

We showed that $NO_2$ data could improve calibration performance in both MLR and RF models. The $NO_2$ data we referred to was measured from an expensive reference monitor and is very reliable. However, it is not always feasible to have an $NO_2$ instrument with high accuracy collocated with a low-cost PM sensor. An alternatives is to use low-cost $NO_2$ sensors. However, their performance remains questionable. To solve this issue, we investigated the effectiveness of using $NO_2$ measurements collected from distant reliable $NO_2$ monitoring sites, called distant $NO_2$, whose location is not that far from a low-cost $PM_{2.5}$ sensor. It was demonstrated that distant $NO_2$ is effective for calibration models based on the MLR and RF algorithms when distant $NO_2$ has a high correlation with collocated $NO_2$. Furthermore, we showed that MLR method can achieve a similar calibration performance to the RF method when reliable distant $NO_2$ is considered.

We performed an evaluation of different PA-II units and found that incorporating $NO_2$ significantly enhanced calibration performance across different PA-II units. This consistency held even when using models trained with different sensors at the same location, reinforcing the reliability of generating consistent data across these units. Additionally, the uncertainty analysis underscored a substantial performance boost by including $NO_2$ in the MLR method, showing a marked difference compared to its omission.

*Data availability.* All data can be provided by the authors upon request.

*Author contributions.* TEXT

*Competing interests.* The authors declare that they have no conflict of interest.

*Acknowledgements.* This paper is financially supported by the Ministry of Trade, Industry and Energy(MOTIE, Korea) through the fostering project of 'Smart City Urban Infrastructure Air Quality Real-time Monitoring and Prediction Platform Technology Development ' supervised by the Korea Institute for Advancement of Technology(KIAT).

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

**Table 1.** Information about 14 PA-II units, such as their ID, location (latitude and longitude), sensor name, start time of measurement, end time of measurement, and non-operating months.

| ID | Latitude | Longitude | Sensor Name | Start Time of Measurement | End Time of Measurement | Non-Operating Months |
|---|---|---|---|---|---|---|
| 1866 | 33.999978 | -117.41676 | RIVR_Co-loc1 | 7/10/17 | 4/27/20 | Sep., Oct., Nov., and Dec. 2018 |
| 1854 | 33.999503 | -117.41602 | RIVR_Co-loc2 | 7/10/17 | 4/27/20 | |
| 2346 | 33.999978 | -117.41676 | RIVR_Co-loc3 | 7/31/17 | 4/27/20 | |
| 2325 | 33.999978 | -117.41676 | RIVR_Co-loc4 | 7/31/17 | 4/27/20 | Sep., Oct., Nov., and Dec. 2018 |
| 2167 | 33.999978 | -117.41676 | RIVR_Co-loc5 | 7/17/17 | 4/27/20 | |
| 2155 | 33.999978 | -117.41676 | RIVR_Co-loc6 | 7/17/17 | 4/27/20 | May, 2018 |
| 2612 | 33.999515 | -117.41595 | RIVR_Co-loc7 | 8/7/17 | 4/27/20 | |
| 2758 | 33.999978 | -117.41676 | RIVR_Co-loc8 | 8/11/17 | 4/27/20 | Sep. 2018 |
| 3537 | 33.999381 | -117.41601 | RIVR_Co-loc9 | 9/20/17 | 4/27/20 | May, Sep., Oct., Nov., and Dec. 2018 |
| 4748 | 33.999516 | -117.41594 | RIVR_Co-loc10 | 11/22/17 | 4/27/20 | May, Aug., Sep., Oct., Nov., and Dec. 2018 and Jan 2019 |
| 4731 | 33.999504 | -117.41593 | RIVR_Co-loc11 | 11/22/17 | 3/1/19 | Jan., Feb., Mar., Sep., Oct., Nov., and Dec. 2018 |
| 5280 | 33.99946 | -117.41594 | RIVR_Co-loc12 | 12/12/17 | 4/27/20 | May, Sep., Oct., Nov., and Dec. 2018 |
| 5284 | 33.999451 | -117.41591 | RIVR_Co-loc13 | 12/12/17 | 4/27/20 | May, Sep., Oct., Nov., and Dec. 2018 |
| 6806 | 33.999583 | -117.41621 | RIVR_Co-loc14 | 1/30/18 | 4/27/20 | Apr., Sep., Oct., and Nov. 2018 |
| 6912 | 33.999482 | -117.41627 | RIVR_Co-loc15 | 1/31/18 | 4/27/20 | Apr., Sep., Oct., and Nov. 2018 |
| 9226 | 33.999389 | -117.41633 | RIVR_Co-loc16 | 3/24/18 | 4/27/20 | Apr., Sep., Oct., Nov., and Dec. 2018 |
| 9358 | 33.999319 | -117.41638 | RIVR_Co-loc17 | 3/25/18 | 4/27/20 | Apr., Sep., Oct., Nov., and Dec. 2018 |

**Table 2.** $R^2$, RMSE, and MAE of the PA-II unit against the BAM-1020 based on the hourly PM2.5 measurement data for each month.

| | Jan-18 | Feb-18 | Mar-18 | Apr-18 | May-18 | Jun-18 | Jul-18 | Aug-18 | Sep-18 | Oct-18 | Nov-18 | Dec-18 |
|---|---|---|---|---|---|---|---|---|---|---|---|---|
| $R^2$ | 0.936 | 0.799 | 0.845 | 0.759 | 0.659 | 0.695 | 0.359 | 0.816 | 0.591 | 0.784 | 0.829 | 0.905 |
| RMSE | 4.201 | 3.735 | 2.932 | 3.938 | 3.477 | 4.097 | 5.615 | 3.204 | 4.550 | 3.650 | 3.832 | 3.765 |
| MAE | 3.171 | 2.721 | 2.196 | 3.098 | 2.716 | 3.267 | 3.597 | 2.424 | 3.358 | 2.844 | 2.913 | 2.743 |
| Intercept | 3.898 | 4.229 | 2.898 | 7.090 | 4.694 | 7.925 | 6.721 | 4.692 | 6.357 | 2.682 | 3.269 | 1.445 |
| Slope | 0.502 | 0.475 | 0.525 | 0.446 | 0.486 | 0.475 | 0.434 | 0.459 | 0.382 | 0.420 | 0.409 | 0.472 |

| | Jan-19 | Feb-19 | Mar-19 | Apr-19 | May-19 | Jun-19 | Jul-19 | Aug-19 | Sep-19 | Oct-19 | Nov-19 | Dec-19 |
|---|---|---|---|---|---|---|---|---|---|---|---|---|
| $R^2$ | 0.884 | 0.750 | 0.735 | 0.618 | 0.801 | 0.730 | 0.893 | 0.405 | 0.441 | 0.523 | 0.880 | 0.813 |
| RMSE | 3.326 | 2.940 | 2.753 | 3.703 | 3.146 | 3.403 | 4.127 | 4.220 | 3.292 | 4.768 | 4.474 | 3.866 |
| MAE | 2.485 | 2.216 | 2.124 | 2.892 | 2.349 | 2.700 | 3.082 | 2.564 | 2.558 | 3.360 | 3.238 | 2.934 |
| Intercept | 1.961 | 2.190 | 1.881 | 4.065 | 2.525 | 3.225 | 3.070 | 5.649 | 5.312 | 5.088 | 2.976 | 1.165 |
| Slope | 0.397 | 0.354 | 0.427 | 0.385 | 0.418 | 0.383 | 0.575 | 0.428 | 0.511 | 0.483 | 0.497 | 0.572 |

**Table 3.** A summary statistics of daily and hourly $PM_{2.5}$ measured from an FRM, BAM-1020, and PA-II 7 unit

| | Daily $PM_{2.5}$ | | | Hourly $PM_{2.5}$ | |
|---|---|---|---|---|---|
| | FRM | BAM-1020 | PA-II | BAM-1020 | PA-II |
| Min ($\mu g/m^3$) | 1.2 | 0 | 0.199 | 0 | 0.019 |
| Max ($\mu g/m^3$) | 66.3 | 68.3 | 129.069 | 159 | 263.062 |
| Mean ($\mu g/m^3$) | 11.69 | 12.13 | 18.247 | 12.171 | 18.367 |
| Standard deviation ($\mu g/m^3$) | 6.88 | 9.16 | 13.854 | 9.23 | 17.61 |

**Table 4.** A list of selected feature vectors in MLR methods

| Feature Vector | PM$_{2.5}$ | T | RH | NO$_2$ | PM$_{2.5}\times$T | PM$_{2.5}\times$RH | PM$_{2.5}\times$NO$_2$ | T$\times$RH | T$\times$NO$_2$ | RH$\times$NO$_2$ | PM$_{2.5}\times$T$\times$RH |
|---|---|---|---|---|---|---|---|---|---|---|---|
| 1 | X | | | | | | | | | | |
| 2 | X | X | | | | | | | | | |
| 3 | X | | X | | | | | | | | |
| 4 | X | | X | | | | X | | | | |
| 5 | X | X | X | | | | | | | | |
| 6 | X | X | X | | | | X | | | | |
| 7 | X | X | X | | | | X | | X | | |
| 8 | X | X | X | | | X | X | | X | | |
| 9 | X | X | X | | | X | X | | X | | X |
| 10 | X | | | X | | | | | | | |
| 11 | X | | | X | | | | X | | | |
| 12 | X | X | | X | | | | | | | |
| 13 | X | X | | X | | | | X | | | |
| 14 | X | X | | X | X | | | X | | | |
| 15 | X | X | | X | X | | | X | | X | |
| 16 | X | X | X | X | | | | | | | |
| 17 | X | X | X | X | | | X | | | | |
| 18 | X | X | X | X | | | X | | X | | |
| 19 | X | X | X | X | X | | X | | X | | |
| 20 | X | X | X | X | X | | X | | X | X | |
| 21 | X | X | X | X | X | | X | X | X | X | |

**Table 5.** Calibration result ($R^2$, RMSE ($\mu g/m^3$), and MAE ($\mu g/m^3$)) of hourly PM$_{2.5}$ concentrations using MLR for the PA-II 7 unit based on the selected combinations.

| | NO$_2$ not included | | | | | | | NO$_2$ included | | | | | |
| | Training Set | | | Test Set | | | | Training Set | | | Test Set | | |
| Feature Vector | $R^2$ | RMSE | MAE | $R^2$ | RMSE | MAE | Feature Vector | $R^2$ | RMSE | MAE | $R^2$ | RMSE | MAE |
|---|---|---|---|---|---|---|---|---|---|---|---|---|---|
| 1 | 0.803 | 4.272 | 3.279 | 0.731 | 4.513 | 3.418 | 10 | 0.806 | 4.241 | 3.259 | 0.741 | 4.424 | 3.329 |
| 2 | 0.814 | 4.150 | 3.185 | 0.755 | 4.305 | 3.194 | 11 | 0.806 | 4.236 | 3.255 | 0.741 | 4.423 | 3.326 |
| 3 | 0.813 | 4.160 | 3.203 | 0.760 | 4.263 | 3.165 | 12 | 0.826 | 4.010 | 3.075 | 0.789 | 3.997 | 2.871 |
| 4 | 0.820 | 4.087 | 3.109 | 0.763 | 4.232 | 3.132 | 13 | 0.827 | 3.997 | 3.071 | 0.789 | 3.993 | 2.857 |
| 5 | 0.816 | 4.125 | 3.174 | 0.763 | 4.234 | 3.129 | 14 | 0.829 | 3.977 | 3.042 | 0.792 | 3.962 | 2.843 |
| 6 | 0.821 | 4.069 | 3.093 | 0.765 | 4.211 | 3.100 | 15 | 0.829 | 3.975 | 3.041 | 0.793 | 3.954 | 2.838 |
| 7 | 0.822 | 4.054 | 3.098 | 0.772 | 4.154 | 3.043 | 16 | 0.826 | 4.008 | 3.077 | 0.790 | 3.986 | 2.866 |
| 8 | 0.824 | 4.040 | 3.086 | 0.772 | 4.151 | 3.023 | 17 | 0.829 | 3.979 | 3.028 | 0.789 | 3.990 | 2.863 |
| 9 | 0.825 | 4.022 | 3.075 | 0.771 | 4.161 | 3.012 | 18 | 0.831 | 3.958 | 3.029 | 0.798 | 3.912 | 2.793 |
| | | | | | | | 19 | 0.831 | 3.950 | 3.026 | 0.796 | 3.925 | 2.790 |
| | | | | | | | 20 | 0.832 | 3.945 | 3.025 | 0.797 | 3.920 | 2.782 |
| | | | | | | | 21 | 0.832 | 3.941 | 3.019 | 0.797 | 3.913 | 2.777 |

**Table 6.** A list of selected feature vectors in RF methods

| Feature Vector | $PM_{2.5}$ | T | RH | $NO_2$ | $PM_{2.5}\times T$ | $PM_{2.5}\times RH$ | $PM_{2.5}\times NO_2$ | $T\times RH$ | $T\times NO_2$ | $RH\times NO_2$ | $PM_{2.5}\times T\times RH$ |
|---|---|---|---|---|---|---|---|---|---|---|---|
| 1 | X | | | | | | | | | | |
| 2 | X | X | | | | | | | | | |
| 3 | X | | X | | | | | | | | |
| 4 | X | | X | | | | X | | | | |
| 5 | X | X | X | | | | | | | | |
| 6 | X | X | X | | | | | | X | | |
| 7 | X | X | X | | | X | | | X | | |
| 8 | X | X | X | | | X | X | | X | | |
| 9 | X | X | X | | | X | X | | X | | X |
| 10 | X | | | X | | | | | | | |
| 11 | X | | | X | | | | X | | | |
| 12 | X | | X | X | | | | | | | |
| 13 | X | X | | X | X | | | | | | |
| 14 | X | X | | X | | | | X | | X | |
| 15 | X | X | | X | X | | | X | | X | |
| 16 | X | X | X | X | | | | | | | |
| 17 | X | X | X | X | X | | | | | | |
| 18 | X | X | X | X | | | X | | X | | |
| 19 | X | X | X | X | X | | | | | X | X |
| 20 | X | X | X | X | X | | | X | | X | X |
| 21 | X | X | X | X | X | | | X | X | X | X |
| 22 | X | X | X | X | X | | X | X | X | X | X |

**Table 7.** Calibration result ($R^2$, RMSE ($\mu g/m^3$), and MAE ($\mu g/m^3$)) of hourly $PM_{2.5}$ concentrations using RF for for the PA-II 7 unit based on the selected combinations.

| | NO₂ not included | | | | | | | NO₂ included | | | | | |
| | Training Set | | | Test Set | | | | Training Set | | | Test Set | | |
| Feature Vector | $R^2$ | RMSE | MAE | $R^2$ | RMSE | MAE | Feature Vector | $R^2$ | RMSE | MAE | $R^2$ | RMSE | MAE |
|---|---|---|---|---|---|---|---|---|---|---|---|---|---|
| 1 | 0.826 | 4.014 | 3.072 | 0.739 | 4.439 | 3.318 | 10 | 0.820 | 4.080 | 3.116 | 0.740 | 4.434 | 3.300 |
| 2 | 0.842 | 3.830 | 2.933 | 0.764 | 4.223 | 3.156 | 11 | 0.821 | 4.074 | 3.109 | 0.738 | 4.451 | 3.305 |
| 3 | 0.857 | 3.632 | 2.785 | 0.786 | 4.026 | 2.951 | 12 | 0.861 | 3.588 | 2.748 | 0.791 | 3.972 | 2.925 |
| 4 | 0.875 | 3.398 | 2.611 | 0.786 | 4.024 | 2.957 | 13 | 0.885 | 3.269 | 2.522 | 0.794 | 3.945 | 2.861 |
| 5 | 0.883 | 3.290 | 2.526 | 0.785 | 4.034 | 2.970 | 14 | 0.885 | 3.262 | 2.519 | 0.797 | 3.918 | 2.887 |
| 6 | 0.862 | 3.568 | 2.740 | 0.787 | 4.014 | 2.955 | 15 | 0.886 | 3.250 | 2.505 | 0.793 | 3.957 | 2.875 |
| 7 | 0.884 | 3.276 | 2.515 | 0.779 | 4.092 | 2.964 | 16 | 0.893 | 3.154 | 2.427 | 0.805 | 3.842 | 2.836 |
| 8 | 0.861 | 3.584 | 2.747 | 0.782 | 4.059 | 2.956 | 17 | 0.920 | 2.720 | 2.092 | 0.797 | 3.918 | 2.840 |
| 9 | 0.905 | 2.968 | 2.257 | 0.785 | 4.029 | 2.853 | 18 | 0.920 | 2.722 | 2.095 | 0.805 | 3.840 | 2.831 |
| | | | | | | | 19 | 0.921 | 2.706 | 2.080 | 0.794 | 3.942 | 2.860 |
| | | | | | | | 20 | 0.921 | 2.699 | 2.073 | 0.795 | 3.936 | 2.857 |
| | | | | | | | 21 | 0.894 | 3.130 | 2.401 | 0.794 | 3.946 | 2.856 |
| | | | | | | | 22 | 0.915 | 2.800 | 2.121 | 0.798 | 3.912 | 2.850 |

**Table 8.** Calibration result ($R^2$, RMSE (µg/m³), and MAE (µg/m³)) of hourly PM$_{2.5}$ concentrations using MLR and RF models for the PA-II 7 unit based on the selected combinations additionally with distant NO$_2$.

| Site ID | Feature Vector | MLR collocated NO$_2$ | | | MLR Distant NO$_2$ | | | RF collocated NO$_2$ | | | RF Distant NO$_2$ | | |
|---|---|---|---|---|---|---|---|---|---|---|---|---|---|
| | | $R^2$ | RMSE | MAE | $R^2$ | RMSE | MAE | $R^2$ | RMSE | MAE | $R^2$ | RMSE | MAE |
| 06-065-8005 | 10 | 0.741 | 4.424 | 3.329 | 0.742 | 4.417 | 3.320 | 0.740 | 4.434 | 3.300 | 0.739 | 4.442 | 3.304 |
| | 11 | 0.741 | 4.423 | 3.326 | 0.743 | 4.411 | 3.311 | 0.738 | 4.451 | 3.305 | 0.738 | 4.454 | 3.306 |
| | 12 | 0.789 | 3.997 | 2.871 | 0.786 | 4.018 | 2.879 | 0.791 | 3.972 | 2.925 | 0.790 | 3.983 | 2.934 |
| | 13 | 0.789 | 3.993 | 2.857 | 0.787 | 4.011 | 2.861 | 0.794 | 3.945 | 2.861 | 0.789 | 3.994 | 2.902 |
| | 14 | 0.792 | 3.962 | 2.843 | 0.791 | 3.978 | 2.842 | 0.797 | 3.918 | 2.887 | 0.791 | 3.970 | 2.923 |
| | 15 | 0.793 | 3.954 | 2.838 | 0.791 | 3.978 | 2.844 | 0.793 | 3.957 | 2.875 | 0.787 | 4.017 | 2.917 |
| | 16 | 0.790 | 3.986 | 2.866 | 0.787 | 4.009 | 2.875 | 0.805 | 3.842 | 2.836 | 0.802 | 3.873 | 2.854 |
| | 17 | 0.789 | 3.990 | 2.863 | 0.787 | 4.011 | 2.870 | 0.797 | 3.918 | 2.840 | 0.793 | 3.951 | 2.860 |
| | 18 | 0.798 | 3.912 | 2.793 | 0.795 | 3.936 | 2.803 | 0.805 | 3.840 | 2.831 | 0.802 | 3.870 | 2.848 |
| | 19 | 0.796 | 3.925 | 2.790 | 0.794 | 3.950 | 2.800 | 0.794 | 3.942 | 2.860 | 0.790 | 3.983 | 2.884 |
| | 20 | 0.797 | 3.920 | 2.782 | 0.795 | 3.933 | 2.780 | 0.795 | 3.936 | 2.857 | 0.791 | 3.978 | 2.877 |
| | 21 | 0.797 | 3.913 | 2.777 | 0.796 | 3.931 | 2.777 | 0.794 | 3.946 | 2.856 | 0.790 | 3.986 | 2.879 |
| | | | | | | | | 0.798 | 3.912 | 2.850 | 0.794 | 3.946 | 2.865 |
| 06-071-0027 | 10 | 0.741 | 4.424 | 3.329 | 0.715 | 4.645 | 3.563 | 0.740 | 4.434 | 3.300 | 0.734 | 4.488 | 3.345 |
| | 11 | 0.741 | 4.423 | 3.326 | 0.715 | 4.641 | 3.549 | 0.738 | 4.451 | 3.305 | 0.729 | 4.525 | 3.367 |
| | 12 | 0.789 | 3.997 | 2.871 | 0.694 | 4.807 | 3.739 | 0.791 | 3.972 | 2.925 | 0.781 | 4.069 | 2.994 |
| | 13 | 0.789 | 3.993 | 2.857 | 0.695 | 4.799 | 3.706 | 0.794 | 3.945 | 2.861 | 0.692 | 4.826 | 3.624 |
| | 14 | 0.792 | 3.962 | 2.843 | 0.696 | 4.797 | 3.673 | 0.797 | 3.918 | 2.887 | 0.693 | 4.815 | 3.646 |
| | 15 | 0.793 | 3.954 | 2.838 | 0.682 | 4.906 | 3.778 | 0.793 | 3.957 | 2.875 | 0.689 | 4.850 | 3.648 |
| | 16 | 0.790 | 3.986 | 2.866 | 0.701 | 4.751 | 3.681 | 0.805 | 3.842 | 2.836 | 0.761 | 4.247 | 3.170 |
| | 17 | 0.789 | 3.990 | 2.863 | 0.714 | 4.651 | 3.576 | 0.797 | 3.918 | 2.840 | 0.733 | 4.494 | 3.325 |
| | 18 | 0.798 | 3.912 | 2.793 | 0.720 | 4.602 | 3.531 | 0.805 | 3.840 | 2.831 | 0.746 | 4.381 | 3.289 |
| | 19 | 0.796 | 3.925 | 2.790 | 0.721 | 4.593 | 3.516 | 0.794 | 3.942 | 2.860 | 0.722 | 4.586 | 3.423 |
| | 20 | 0.797 | 3.920 | 2.782 | 0.721 | 4.595 | 3.516 | 0.795 | 3.936 | 2.857 | 0.721 | 4.592 | 3.422 |
| | 21 | 0.797 | 3.913 | 2.777 | 0.702 | 4.746 | 3.669 | 0.794 | 3.946 | 2.856 | 0.744 | 4.401 | 3.256 |
| | | | | | | | | 0.798 | 3.912 | 2.850 | 0.727 | 4.542 | 3.386 |

**Table 9.** Calibration results of hourly PM$_{2.5}$ concentrations measured from the PA-II 8 in 2019 using MLR-based calibration model learned with training data collected from the PA-II 8 in 2018.

| | NO$_2$ not included | | | | | | | NO$_2$ included | | | | | |
| | Training Set | | | Test Set | | | | Training Set | | | Test Set | | |
| Feature Vector | $R^2$ | RMSE | MAE | $R^2$ | RMSE | MAE | Feature Vector | $R^2$ | RMSE | MAE | $R^2$ | RMSE | MAE |
|---|---|---|---|---|---|---|---|---|---|---|---|---|---|
| 1 | 0.786 | 4.312 | 3.304 | 0.731 | 4.559 | 3.468 | 10 | 0.788 | 4.292 | 3.295 | 0.741 | 4.468 | 3.381 |
| 2 | 0.798 | 4.196 | 3.211 | 0.749 | 4.397 | 3.299 | 11 | 0.789 | 4.289 | 3.293 | 0.742 | 4.459 | 3.375 |
| 3 | 0.797 | 4.208 | 3.231 | 0.760 | 4.307 | 3.223 | 12 | 0.809 | 4.079 | 3.127 | 0.783 | 4.087 | 2.982 |
| 4 | 0.803 | 4.142 | 3.147 | 0.763 | 4.277 | 3.191 | 13 | 0.810 | 4.070 | 3.123 | 0.785 | 4.072 | 2.966 |
| 5 | 0.800 | 4.173 | 3.201 | 0.759 | 4.311 | 3.219 | 14 | 0.811 | 4.051 | 3.099 | 0.788 | 4.042 | 2.951 |
| 6 | 0.805 | 4.123 | 3.127 | 0.762 | 4.281 | 3.185 | 15 | 0.811 | 4.050 | 3.099 | 0.788 | 4.040 | 2.950 |
| 7 | 0.806 | 4.111 | 3.134 | 0.767 | 4.242 | 3.143 | 16 | 0.809 | 4.076 | 3.128 | 0.785 | 4.071 | 2.970 |
| 8 | 0.807 | 4.099 | 3.127 | 0.768 | 4.227 | 3.121 | 17 | 0.811 | 4.050 | 3.083 | 0.785 | 4.071 | 2.967 |
| 9 | 0.808 | 4.091 | 3.121 | 0.770 | 4.214 | 3.128 | 18 | 0.813 | 4.033 | 3.087 | 0.791 | 4.015 | 2.915 |
| | | | | | | | 19 | 0.814 | 4.028 | 3.084 | 0.791 | 4.019 | 2.911 |
| | | | | | | | 20 | 0.814 | 4.023 | 3.083 | 0.792 | 4.006 | 2.895 |
| | | | | | | | 21 | 0.814 | 4.021 | 3.081 | 0.792 | 4.002 | 2.892 |

**Table 10.** Calibration results of hourly PM$_{2.5}$ concentrations measured from the PA-II 8 in 2019 using MLR-based calibration model learned with training data collected from the PA-II 7 in 2018.

| | NO$_2$ not included | | | | NO$_2$ included | | |
| | Test Set | | | | Test Set | | |
| Feature Vector | $R^2$ | RMSE | MAE | Feature Vector | $R^2$ | RMSE | MAE |
|---|---|---|---|---|---|---|---|
| 1 | 0.737 | 4.638 | 3.546 | 10 | 0.747 | 4.549 | 3.458 |
| 2 | 0.757 | 4.459 | 3.364 | 11 | 0.748 | 4.538 | 3.446 |
| 3 | 0.763 | 4.400 | 3.322 | 12 | 0.788 | 4.162 | 3.054 |
| 4 | 0.765 | 4.383 | 3.293 | 13 | 0.790 | 4.145 | 3.031 |
| 5 | 0.765 | 4.388 | 3.301 | 14 | 0.794 | 4.104 | 3.003 |
| 6 | 0.766 | 4.373 | 3.275 | 15 | 0.795 | 4.097 | 3.000 |
| 7 | 0.772 | 4.323 | 3.222 | 16 | 0.789 | 4.151 | 3.048 |
| 8 | 0.772 | 4.318 | 3.208 | 17 | 0.789 | 4.158 | 3.050 |
| 9 | 0.774 | 4.301 | 3.208 | 18 | 0.796 | 4.089 | 2.985 |
| | | | | 19 | 0.795 | 4.100 | 2.984 |
| | | | | 20 | 0.795 | 4.095 | 2.974 |
| | | | | 21 | 0.796 | 4.090 | 2.970 |

**Table 11.** Calibration results of hourly $PM_{2.5}$ concentrations measured from the PA-II 8 in 2019 using MLR-based calibration model learned with training data collected from the PA-II 8 in 2018 (Site ID indicates the monitoring sites for distant $NO_2$).

| Site ID | Feature Vector | MLR collocated $NO_2$ $R^2$ | RMSE | MAE | MLR Distant $NO_2$ $R^2$ | RMSE | MAE |
|---|---|---|---|---|---|---|---|
| 06-065-8005 | 10 | 0.741 | 4.468 | 3.381 | 0.742 | 4.458 | 3.371 |
| | 11 | 0.742 | 4.459 | 3.375 | 0.744 | 4.442 | 3.359 |
| | 12 | 0.783 | 4.087 | 2.982 | 0.783 | 4.089 | 2.976 |
| | 13 | 0.785 | 4.072 | 2.966 | 0.786 | 4.066 | 2.951 |
| | 14 | 0.788 | 4.042 | 2.951 | 0.789 | 4.031 | 2.927 |
| | 15 | 0.788 | 4.040 | 2.950 | 0.789 | 4.033 | 2.930 |
| | 16 | 0.785 | 4.071 | 2.970 | 0.785 | 4.075 | 2.966 |
| | 17 | 0.785 | 4.071 | 2.967 | 0.785 | 4.076 | 2.960 |
| | 18 | 0.791 | 4.015 | 2.915 | 0.790 | 4.022 | 2.911 |
| | 19 | 0.791 | 4.019 | 2.911 | 0.790 | 4.026 | 2.908 |
| | 20 | 0.792 | 4.006 | 2.895 | 0.793 | 3.998 | 2.877 |
| | 21 | 0.792 | 4.002 | 2.892 | 0.793 | 3.995 | 2.875 |
| 06-071-0027 | 10 | 0.741 | 4.468 | 3.381 | 0.716 | 4.681 | 3.600 |
| | 11 | 0.742 | 4.459 | 3.375 | 0.716 | 4.680 | 3.591 |
| | 12 | 0.783 | 4.087 | 2.982 | 0.684 | 4.937 | 3.887 |
| | 13 | 0.785 | 4.072 | 2.966 | 0.684 | 4.937 | 3.864 |
| | 14 | 0.788 | 4.042 | 2.951 | 0.680 | 4.965 | 3.850 |
| | 15 | 0.788 | 4.040 | 2.950 | 0.672 | 5.030 | 3.914 |
| | 16 | 0.785 | 4.071 | 2.970 | 0.693 | 4.870 | 3.816 |
| | 17 | 0.785 | 4.071 | 2.967 | 0.706 | 4.764 | 3.704 |
| | 18 | 0.791 | 4.015 | 2.915 | 0.710 | 4.733 | 3.676 |
| | 19 | 0.791 | 4.019 | 2.911 | 0.713 | 4.705 | 3.646 |
| | 20 | 0.792 | 4.006 | 2.895 | 0.713 | 4.709 | 3.643 |
| | 21 | 0.792 | 4.002 | 2.892 | 0.699 | 4.818 | 3.756 |

**Table 12.** Statistics of uncertainty analysis to selected feature vectors and t-values.

| Feature Vector | Mean of RMSE | Std. Dev. of RMSE | Feature Vector | Mean of RMSE | Std. Dev. of RMSE | t-value | DoF |
|---|---|---|---|---|---|---|---|
| $\{PM_{2.5}\}$ | 4.5095 | 0.1026 | $\{PM_{2.5}, NO_2\}$ | 4.4202 | 0.1037 | 19.3580 | 1,998 |
| $\{PM_{2.5}, T\}$ | 4.3084 | 0.1000 | $\{PM_{2.5}, T, NO_2\}$ | 3.9979 | 0.1173 | 63.7008 | 1,998 |
| $\{PM_{2.5}, RH\}$ | 4.2598 | 0.0995 | $\{PM_{2.5}, RH, NO_2\}$ | 4.1548 | 0.1074 | 22.6792 | 1,998 |
| $\{PM_{2.5}, T, RH\}$ | 4.2387 | 0.1050 | $\{PM_{2.5}, T, RH, NO_2\}$ | 3.9865 | 0.1156 | 51,0686 | 1,998 |

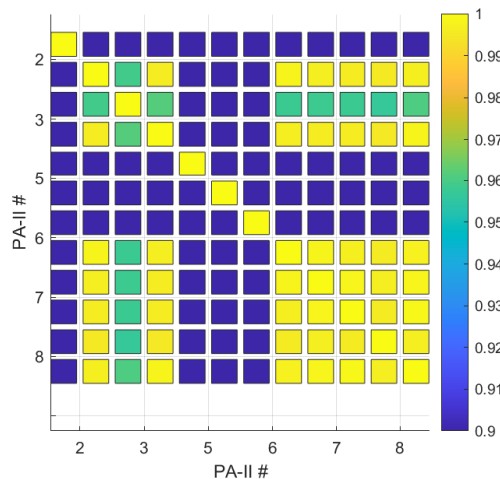

**Figure 1.** Correlation among all PMS 5003 sensors of the selected units PA-II 2, 3, 5, 6, 7, and 8. The left and right of each number on the x-axis represent PMS A and B sensors for its corresponding PA-II unit, respectively.

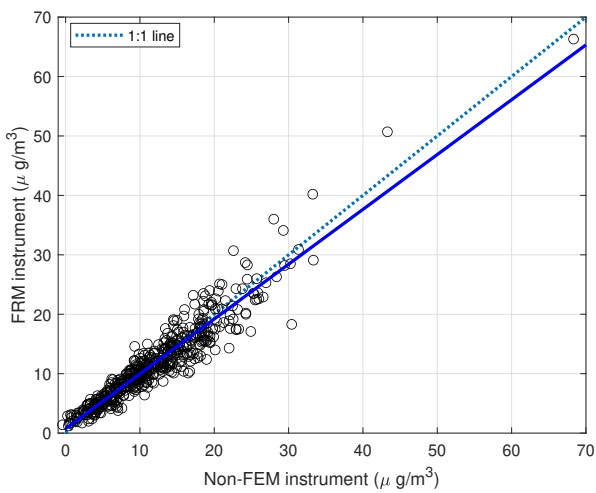

**Figure 2.** Scatter plot for daily PM$_{2.5}$ comparison of BAM-1020 (Non-FEM) instrument with the FRM instrument.

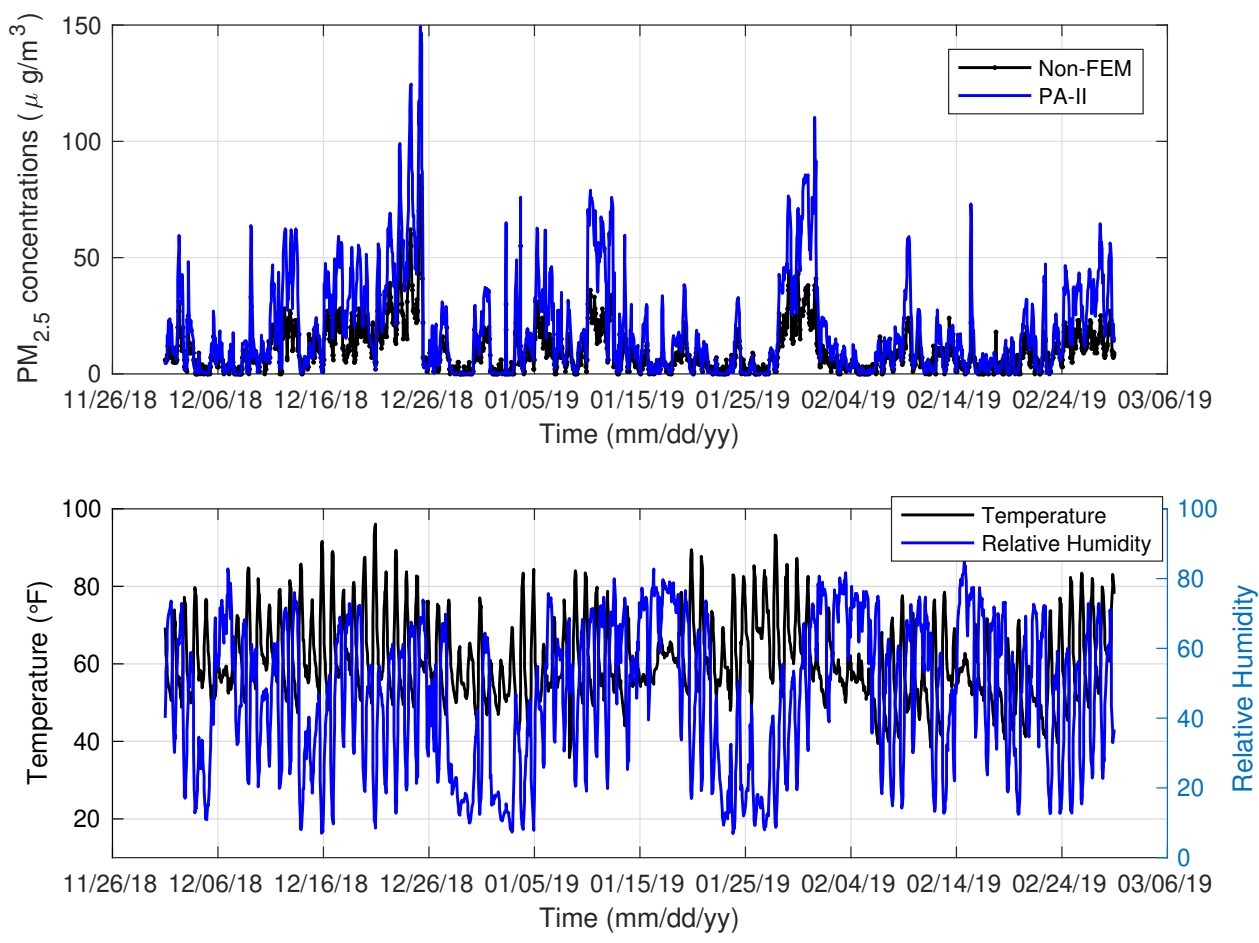

**Figure 3.** Hourly PM$_{2.5}$ concentrations measured from BAM-1020 (Non-FEM) and PA-II 7, and hourly temperature and relative humidity measured from PA-II 7 from Dec. 2018 to Feb. 2019.

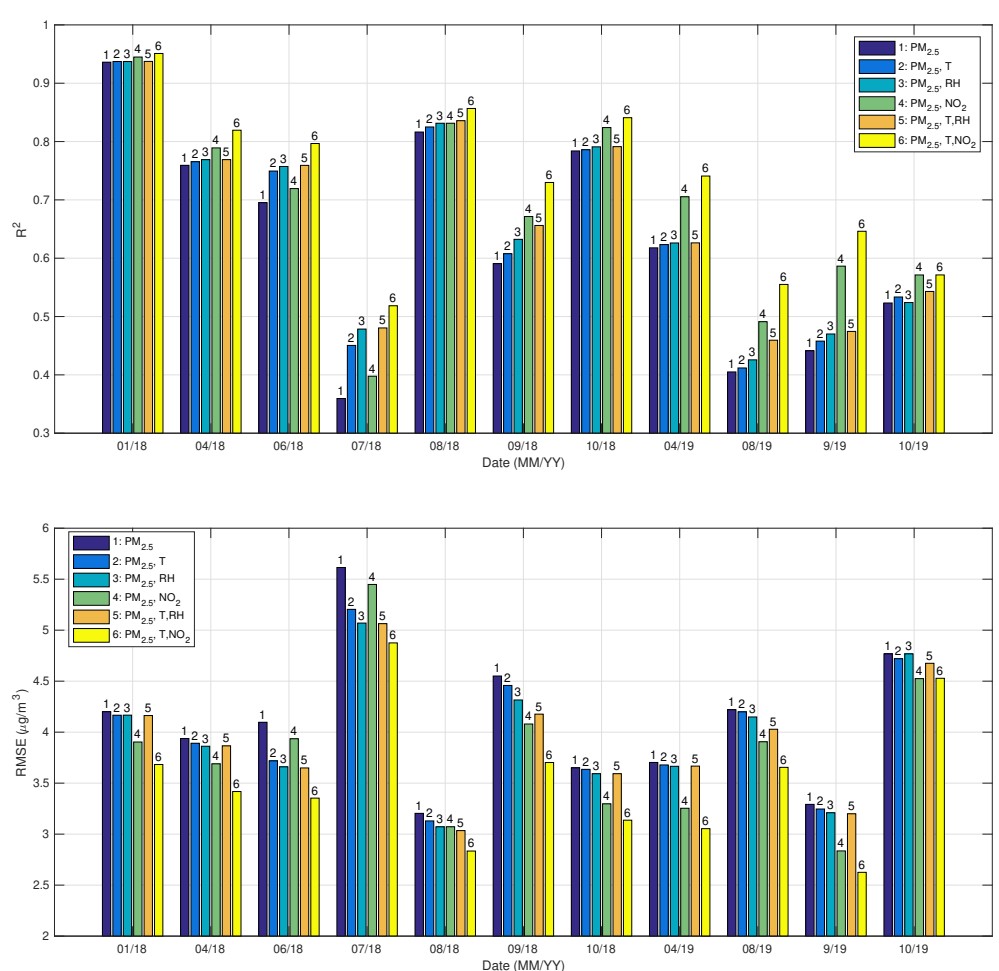

**Figure 4.** $R^2$ and RMSE using MLR method for the PA-II unit with the BAM-1020 for the selected months based on the following feature vectors; 1:($PM_{2.5}$), 2:($PM_{2.5}$, T), 3:($PM_{2.5}$, RH), 4:($PM_{2.5}$, $NO_2$), 5:($PM_{2.5}$, T, RH), and 6:($PM_{2.5}$, T, $NO_2$).