# Peer review of "Evaluation of Calibration Performance of a Low-cost Particulate Matter Sensor Using Collocated and Distant NO2"

_EGUsphere, 2023_

## Referee Comment (RC2)

Reviewer comments for:

**Evaluation of Calibration Performance of a Low-cost Particulate Matter Sensor Using Colocated and Distant NO$_2$**

This review is for the above manuscript submitted for publication in Atmospheric Measurement Techniques. The manuscript uses co-located and regulatory monitoring-based measurements to build calibration models for low-cost PM$_{2.5}$ sensors. The authors test several combinations of four variables such as PM2.5 measurements, temperature, relative humidity, and NO$_2$, where the NO$_2$ measurements are from either collocated or nearby instrumentation. The authors conclude that collocated or nearby NO$_2$ measurements should be used for calibration models since model performance improves in terms of statistical measures such as RMSE, MAE, and R$^2$. However, in its current form, the authors' comparisons are very limited in scope (single sensor), lack appropriate choice of performance parameters (e.g., adjusted R$^2$), show very limited performance improvement (~5%), and are marred by lack of uncertainty analysis and poor presentation of methods and results. **Reviewer 1 has discussed the limitations of a single sensor so I will focus on other aspects. I recommend that the authors significantly revise and resubmit this manuscript for further consideration.**

1. **Very limited in scope and performance improvement:** Given that the authors focused the comparison on calibration of a single sensor, the weight of "substantial contribution" of this manuscript falls on performance improvements of calibration models associated with that sensor. Unfortunately, the improvements on inclusion of NO$_2$ are quite minimal. For example, in Tables 3 and 4, the best performances of models with and without NO$_2$ are ~5% of each other. Does that qualify this work as "represent(ing) a substantial contribution to scientific progress" as is required by AMT? I disagree. **I suggest that the authors conduct the analysis for the excluded sensor (sensor #8) that otherwise passes all checks, but was not included in the analysis for an unknown reason, as also pointed by reviewer 1**.

2. **Choice of performance parameters and lack of uncertainty analysis:** While the authors include three performance measures, despite considering models with multiple and changing number of variables, the authors fail to include the most important one: adjusted R$^2$. The authors have clearly used the multiple R$^2$ squared value to compare model fits; however, multiple R$^2$ will increase on addition of even poorly correlated variables. I suggest that the authors report adjusted R$^2$ results. Additionally, presentation of such calibration results would also benefit from an uncertainty analysis, and a key manuscript cited by the authors uses bootstrapping to do just that (Hua et al., 2021). I strongly recommend uncertainty (in terms of standard deviation) be considered when presenting performance metrics associated with such comparisons. **The authors can then answer the**

**question: are the distributions of performance parameters statistically significantly different with or without NO₂? I would consider answering that question as a significant contribution.**

**3. Poor presentation:** Large sections of the manuscript are unnecessarily detailed, and could be moved into tabular form whether in the main manuscript or the supplement. These include large portion of the lines 198-222 and 233-246 which are two representative examples. Additional examples include lists of variables shown in text format, which is laborious to read or keep track of (e.g., Lines 355-364). Additionally, key details of the authors' methodology such as performance metrics and intercomparison exercises are dispersed throughout the Results section (Sect. 3.1 to 3.6). **I suggest that authors separate the methods portions of these results and discuss them in a separate subsection under Methods called "Instrument intercomparisons".**

Minor comments

1. Lines 123-139 The authors start off with a large dataset but remove data points using some filters. I suggest that the authors add a supplementary table showing how many data points were removed at each step.

2. Lines 412-413 and Lines 287-290 The language used by the authors is unclear. I suggest either expanding on these sentences or rephrasing them so that the point is made clearly.

1. Hua, J., Zhang, Y., de Foy, B., Mei, X., Shang, J., Zhang, Y., Sulaymon, I. D., & Zhou, D. (2021). Improved PM2.5 concentration estimates from low-cost sensors using calibration models categorized by relative humidity. *Aerosol Science and Technology*, *55*(5), 600–613. https://doi.org/10.1080/02786826.2021.1873911

---

## Author Comment (AC1)

**Response to Comments from Editors and Referees**

**Information of previously submitted manuscript**

- Manuscript number: egusphere-2023-1344
- Title: Evaluation of Calibration Performance of a Low-cost Particulate Matter Sensor Using Collocated and Distant $NO_2$
- Authors: Kabseok Ko, Ramesh R. Rao, and Seokheon Cho
- Status: Under Review

The authors would like to thank the referees for their careful reviews and insightful comments. We prepared responses for each comment from the editors and referees and revised our manuscript to reflect feedback.

**Referee #1**

The authors appreciate Referee #1's kind and valuable comments.

[Major Comments]

1. Section 3.1: 14 sensors were originally identified in this study, however only 5 were selected based on their months of valid measurements data. Of these 5, 2 were explicitly eliminated based on correlation analysis between the sensors' and their A and B units. Based on Figure 1 it seems like both PA-II 7 and 8 would be suitable for this study while PA-II 2, 3, 5, & 6 were not (Sensor 5 included in Figure 1 but not on line 188). Your final results will be more applicable if you are able to demonstrate improvements in more than 1 sensor, even if the study period is less than 2 years.

(Response)

We showed a series of results for PA-II 7 in our original manuscript. As the referee suggested, we evaluated PA-II 8's calibration performance under the following three cases:

Case 1: Calibration model is learned with the measurements collected from PA-II 8 in 2018 and calibration performance for the trained model is evaluated using data measured from PA-II 8 in 2019.

Case 2: This is similar to Case 1, except that the calibration model is trained with data measured from PA-II 7 in 2018.

Case 3: The measurement data from PA-II 8 with collocated $NO_2$ concentration in 2018 is used as a training dataset, while the data collected from PA-II 8 with either collocated $NO_2$ or distant $NO_2$ concentration in 2019 is used as a test dataset.

In Case 1, we evaluated the calibration model's performance with a test dataset consisting of measurement data from PA-II 8 in 2019. The calibration model is trained with data collected from the same PA-II 8 in 2018. Table R1 shows the calibration results of the PA-II 8 using a multiple linear regression (MLR) method under two different conditions: with and without $NO_2$. We selected the same feature vectors as defined in the original manuscript. We observed that $NO_2$ can enhance calibration performance because all MLR models using $NO_2$, except

combinations #10 and #11, yield lower errors and larger $R^2$ than those without $NO_2$. This observation aligns with the results shown in Table 3 of the original manuscript.

Additionally, compared to the calibration performance for PA-II 7 shown in Table 3 of the original manuscript, PA-II 8 shows slightly larger RMSE and MAE, but similar $R^2$.

**Table R1. Calibration results of hourly PM$_{2.5}$ concentrations measured from the PA-II 8 in 2019 using MLR-based calibration model learned with training data collected from the PA-II 8 in 2018.**

| NO$_2$ not included | | | | NO$_2$ included (i.e., collocated NO$_2$) | | | |
|---|---|---|---|---|---|---|---|
| Feature Vector | $R^2$ | RMSE | MAE | Feature Vector | $R^2$ | RMSE | MAE |
| 1 | 0.731 | 4.559 | 3.468 | 10 | 0.741 | 4.468 | 3.381 |
| 2 | 0.749 | 4.397 | 3.299 | 11 | 0.742 | 4.459 | 3.375 |
| 3 | 0.760 | 4.307 | 3.223 | 12 | 0.783 | 4.087 | 2.982 |
| 4 | 0.763 | 4.277 | 3.191 | 13 | 0.785 | 4.072 | 2.966 |
| 5 | 0.759 | 4.311 | 3.219 | 14 | 0.788 | 4.042 | 2.951 |
| 6 | 0.762 | 4.281 | 3.185 | 15 | 0.788 | 4.040 | 2.950 |
| 7 | 0.767 | 4.242 | 3.143 | 16 | 0.785 | 4.071 | 2.970 |
| 8 | 0.768 | 4.227 | 3.121 | 17 | 0.785 | 4.071 | 2.967 |
| 9 | 0.770 | 4.214 | 3.128 | 18 | 0.791 | 4.015 | 2.915 |
| | | | | 19 | 0.791 | 4.019 | 2.911 |
| | | | | 20 | 0.792 | 4.006 | 2.895 |
| | | | | 21 | 0.792 | 4.002 | 2.892 |

In Case 2, we evaluated the calibration model's performance using a training dataset collected from PA-II 7 in 2018, and a test dataset collected from PA-II 8 in 2019. Table R2 shows calibration results for PA-II 8 using the MLR method under two different conditions, such as with and without $NO_2$. As with the observation in Table R1, $NO_2$ is the key factor enhancing calibration performance. With the exceptions of #10 and 11, all MLR models using $NO_2$ yield lower errors and larger $R^2$ than those without $NO_2$. It is important to compare this result with that shown in Table 3 of the original manuscript, as we used different test datasets. It could be expected that the much worse performance for all feature combinations listed in Table R2 is achieved than for every corresponding feature vector in Table 3 of original manuscript, since the calibration model considered in Table R2 is tested with the data measured from the PA-II 8, whereas it is trained with the measurement data collected from the PA-II 7. $R^2$ values of all feature vectors in Table 10 are similar to those for each corresponding feature vector in Table 5. Unlike $R^2$, we observe larger RMSE and MAE when we populate the training dataset with measurements from PA-II 8 rather than PA-II 7. The maximum differences of RMSE and MAE for each feature vector in Tables 10 and 5 are 0.177 $\mu g/m^3$ and 0.196 $\mu g/m^3$, respectively.

The results shown in Tables R1 and R2 support our conclusion that reliable and consistent PA-II units, which contain two PMS 5003 sensors with high correlation, demonstrate similar calibration performance. This implies that a proposed calibration method can be applied to reliable and consistent PA-II units generally.

**Table R2. Calibration results of hourly PM$_{2.5}$ concentrations measured from PA-II 8 in 2019 using MLR-based calibration model learned with training data collected from PA-II 7 in 2018.**

| NO$_2$ not included | | | | NO$_2$ included (i.e., collocated NO$_2$) | | | |
|---|---|---|---|---|---|---|---|
| Feature Vector | R$^2$ | RMSE | MAE | Feature Vector | R$^2$ | RMSE | MAE |
| 1 | 0.737 | 4.638 | 3.546 | 10 | 0.747 | 4.549 | 3.458 |
| 2 | 0.757 | 4.459 | 3.364 | 11 | 0.748 | 4.538 | 3.446 |
| 3 | 0.763 | 4.400 | 3.322 | 12 | 0.788 | 4.162 | 3.054 |
| 4 | 0.765 | 4.383 | 3.293 | 13 | 0.790 | 4.145 | 3.031 |
| 5 | 0.765 | 4.388 | 3.301 | 14 | 0.794 | 4.104 | 3.003 |
| 6 | 0.766 | 4.373 | 3.275 | 15 | 0.795 | 4.097 | 3.000 |
| 7 | 0.772 | 4.323 | 3.222 | 16 | 0.789 | 4.151 | 3.048 |
| 8 | 0.772 | 4.318 | 3.208 | 17 | 0.789 | 4.158 | 3.050 |
| 9 | 0.774 | 4.301 | 3.208 | 18 | 0.796 | 4.089 | 2.985 |
| | | | | 19 | 0.795 | 4.100 | 2.984 |
| | | | | 20 | 0.795 | 4.095 | 2.974 |
| | | | | 21 | 0.796 | 4.090 | 2.970 |

Lastly, in Case 3, we evaluated the effect of collocated and distant NO$_2$ on the PA-II 8 unit's calibration performance. Table R3 shows the results of MLR-based calibration model for the PA-II 8 when it is verified with the test data considering either collocated or distant NO$_2$. As we explained in the original manuscript, we considered two monitoring sites measuring NO$_2$ near the Rubidoux site. One site (ID 06-065-8005) had NO$_2$ measurements that were much more highly correlated with the Rubidoux site than those from the other site (ID 06-071-00247). We refer to the NO$_2$ concentrations measured from these two sites as "distant NO$_2$." Three columns, describing the values of R$^2$, RMSE, and MAE, of collocated NO$_2$ in Table R3 are exactly the same as those of NO$_2$ included (i.e., collocated NO$_2$) in Table R1. In the case of site 06-065-8005 with high correlation to the Rubidoux site, the consideration of the distant NO$_2$ facilitates improvement of the calibration performance, since all MLR-based calibration models using distant NO$_2$, except combinations #10 and 11, produce lower errors and higher R$^2$ than those without NO$_2$. This result is similar to when we consider the collocated NO$_2$. However, we observe that adding distant NO$_2$ to the test dataset, which is not highly correlated to the NO$_2$ measurement from the reference site, deteriorates the calibration performance. This

is likely because all combinations from #10 to 21 yield lower $R^2$ and greater errors than all combinations excluding $NO_2$, as shown in Table R1. This result is the same as our observation of the PA-II 7 unit's calibration results in Table 5 of the original manuscript.

Hence, these results we draw from Table R3 support the same conclusions we drew from Tables R1 and R2. Reliable and consistent PA-II units achieve similar calibration performance, and our proposed calibration model can be applied to these units generally.

**Table R3. Calibration results of hourly PM$_{2.5}$ concentrations taken from PA-II 8 in 2019 using an MLR-based calibration model learned with data collected from 
[revised manuscript text omitted]

| 1 | 0.738 | 4.447 | 3.193 | 0.750 | 4.344 | 3.204 | 0.729 | 4.524 | 3.436 | 0.731 | 4.513 | 3.418 |
| 2 | 0.746 | 4.382 | 3.132 | 0.775 | 4.125 | 2.925 | 0.763 | 4.231 | 3.110 | 0.755 | 4.305 | 3.194 |
| 3 | 0.752 | 4.329 | 3.094 | 0.776 | 4.111 | 2.971 | 0.761 | 4.253 | 3.163 | 0.760 | 4.263 | 3.165 |

| | | | | | | | | | | | | |
|---|---|---|---|---|---|---|---|---|---|---|---|---|
| 4 | 0.757 | 4.286 | 3.051 | 0.779 | 4.083 | 2.930 | 0.762 | 4.237 | 3.136 | 0.763 | 4.232 | 3.132 |
| 5 | 0.743 | 4.411 | 3.171 | 0.779 | 4.090 | 2.906 | 0.767 | 4.193 | 3.080 | 0.763 | 4.234 | 3.129 |
| 6 | 0.745 | 4.393 | 3.151 | 0.782 | 4.061 | 2.869 | 0.769 | 4.182 | 3.056 | 0.765 | 4.211 | 3.100 |
| 7 | 0.749 | 4.355 | 3.127 | 0.787 | 4.012 | 2.828 | 0.779 | 4.091 | 2.964 | 0.772 | 4.154 | 3.043 |
| 8 | 0.748 | 4.364 | 3.073 | 0.786 | 4.020 | 2.823 | 0.779 | 4.087 | 2.950 | 0.772 | 4.151 | 3.023 |
| 9 | 0.742 | 4.420 | 3.058 | 0.783 | 4.051 | 2.798 | 0.778 | 4.101 | 2.940 | 0.771 | 4.161 | 3.012 |
| 10 | 0.735 | 4.477 | 3.181 | 0.762 | 4.243 | 3.100 | 0.736 | 4.470 | 3.383 | 0.741 | 4.424 | 3.329 |
| 11 | 0.721 | 4.592 | 3.208 | 0.762 | 4.246 | 3.101 | 0.736 | 4.468 | 3.382 | 0.741 | 4.423 | 3.326 |
| 12 | 0.774 | 4.133 | 2.882 | 0.806 | 3.832 | 2.651 | 0.794 | 3.946 | 2.805 | 0.789 | 3.997 | 2.871 |
| 13 | 0.762 | 4.244 | 2.880 | 0.806 | 3.832 | 2.639 | 0.794 | 3.946 | 2.806 | 0.789 | 3.993 | 2.857 |
| 14 | 0.764 | 4.226 | 2.874 | 0.811 | 3.782 | 2.579 | 0.798 | 3.908 | 2.787 | 0.792 | 3.962 | 2.843 |
| 15 | 0.762 | 4.241 | 2.888 | 0.811 | 3.775 | 2.572 | 0.799 | 3.900 | 2.783 | 0.793 | 3.954 | 2.838 |
| 16 | 0.769 | 4.178 | 2.949 | 0.805 | 3.842 | 2.646 | 0.794 | 3.950 | 2.805 | 0.790 | 3.986 | 2.866 |
| 17 | 0.769 | 4.175 | 2.944 | 0.806 | 3.833 | 2.631 | 0.793 | 3.960 | 2.804 | 0.789 | 3.990 | 2.863 |
| 18 | 0.775 | 4.126 | 2.919 | 0.807 | 3.821 | 2.663 | 0.805 | 3.840 | 2.693 | 0.798 | 3.912 | 2.793 |
| 19 | 0.770 | 4.168 | 2.900 | 0.806 | 3.831 | 2.668 | 0.805 | 3.841 | 2.693 | 0.796 | 3.925 | 2.790 |
| 20 | 0.770 | 4.170 | 2.904 | 0.803 | 3.858 | 2.690 | 0.805 | 3.835 | 2.687 | 0.797 | 3.920 | 2.782 |
| 21 | 0.765 | 4.218 | 2.936 | 0.803 | 3.860 | 2.692 | 0.805 | 3.837 | 2.690 | 0.797 | 3.913 | 2.777 |

2. Line 298: What is the reasoning behind this 1:1 data split, specifically using the whole year of 2018 to train the models and apply to 2019. This implies that in practice you have to wait a whole year before collecting valid/corrected data with this method which hinders the use of low-cost sensors. And assuming minimal sensor drift from 2018 to 2019 and similar environmental conditions.

(Response)

The 1:1 data split reflects seasonal patterns in $PM_{2.5}$ and other environmental parameters, such as temperature and relative humidity. To more accurately gauge the relationship between a PA-II unit and regulatory measurements across seasons, we used whole-year data for training. To support our efforts, we studied the training period's effect on calibration performance. As shown in Table S4, training with a shorter period like 3 months yields lower RMSE and MAE than training with 6, 9, or 12-month periods. Hence, it is necessary to train calibration models with data collected over a long enough period to fully account for seasonality and provide reliable performance.

Over time, the degradation of electrical components or dust accumulation can cause drift in low-cost PM sensors. Figure R1 shows $PM_{2.5}$ concentrations obtained from both PA-II 7 and 8 units whose internal PMS 5003 sensors have high correlation with each other. Both PA-II units render similar $PM_{2.5}$ concentrations over time, which makes it challenging to verify the amount of drift experienced by each unit. Therefore, we assume that each PA-II unit has different and minimal drift. Under this assumption, when we compare the performance of the two calibration models, which are trained with distinct datasets from the PA-II 7 and PA-II 8 units in 2018, respectively, and verified with the same test dataset collected from PA-II 8 in 2019, minimal drift has a minor effect on calibration performance, since both units demonstrate similar calibration performance through RMSE and MAE. This comparison was described and explained in Tables 9 and 10.

[Figure]

**Figure R1. Results of PM$_{2.5}$ concentrations from both PA-II 7 and 8 units**

[Minor Comments]

1. Figure 1: Please include info about PA sensors A and B in the caption as you did on line 193.

(Response)

We updated the caption for Figure 1 as follows:

Correlation among all PMS 5003 sensors of the selected units PA-II 2, 3, 5, 6, 7, and 8. The left and right of each number on the x-axis represent PMS A and B sensors for its corresponding PA-II unit, respectively.

2. Figure 2: Include a 1:1 line for comparison.

(Response)

We added a 1:1 line to Figure 1.

3. Figure 3: Ensure x-axes are the same for the PM2.5 graph and temperature+RH graph. Figure sizes could be increased to improve readability.

(Response)

We modified the x-axes in the two subfigures to reflect this suggestion.

4. Line 36: Please clarify that FRM and FEM are US EPA designations and may not be applicable to every county.

(Response)

We updated the sentence on line 36 as follows:

The monitoring stations use instruments based on Federal Reference Methods (FRMs) or Federal Equivalent Methods (FEMs), which promote high precision and accuracy. The U.S. Environmental Protection Agency (EPA) approves both FRMs and FEMs as official designations for measuring ambient concentrations. Furthermore, the U.S. EPA carries out various cooperative programs, including those on ambient monitoring methods and technologies, with many other countries in the world.

5. Line 61: "good a correlation" Please correct to "a good correlation".

(Response)

We modified our text as recommended.

6. Line 74: More discussion needed on how NO2 contributes to PM2.5 formation.

(Response)

We updated the sentence on line 74 as follows:

In addition to these direct factors, we examine the impact of the precursor gas $NO_2$, acting as a source of $PM_{2.5}$ emissions, on calibration performance in low-cost $PM_{2.5}$ sensors. In general, $PM_{2.5}$ arises by secondary formation from a chemical reaction between precursor gases,

such as $NO_2$, in the atmosphere some distance downwind from the original emission source (Hodan et al., 2004).

{Reference}: Hodan, W.H. and Barnard, W.R.: Evaluating the Contribution of PM2.5 Precursor Gases and Re-entrained Road Emissions to Mobile Source PM2.5 Particulate Matter Emissions, MACTEC Federal Programs, 2004.

7. Line 127: Typo for US EPA

(Response)
We modified it as recommended.

8. Line 131: What is the purpose of the 2-minute vs 80 sec interval?

(Response)
We updated the sentence on line 131 as follows:

In the first step, when we calculate 1-hour averages of $PM_{2.5}$ measurements generated with 2 min (or 80 sec) intervals, we remove the 1-hour average if the number of $PM_{2.5}$ measurements is less than 27 (or 40). We considered two different measurement intervals for a PA-II unit because its old interval had been 80 sec until May 30, 2019. Its current interval is 2 min.

9. Line 178: Please clarify the difference between the FRM instrument and the BAM instrument. Does the FRM only report daily values?

(Response)
We updated the sentence on line 178 as follows:

The monitoring site we considered has an FRM instrument and a BAM-1020 instrument with the parameter of 88502. These instruments produce daily and hourly $PM_{2.5}$ measurement data, respectively. Since we measure the PA-II units at intervals much shorter than a full day, it is much more reasonable to compare the $PM_{2.5}$ measurement of PA-II units with that of a

BAM-1020 instrument with a shorter measurement interval, rather than an FRM instrument for evaluating the accurate calibration performance of PA-II units. However, we face the limitation that a BAM-1020 instrument can be classified as a non-FEM-compliant device. Therefore, our approach for analyzing PA-II units to appropriately resolve these issues is as follows: we compared the BAM-1020 instrument's readings with daily $PM_{2.5}$ concentrations collected from an FRM instrument to ensure the BAM-1020 provides an acceptable level of performance as an FRM instrument, which is enough to assess the calibration performance of PA-II units. According to this affirmative observation, the BAM-1020 instrument can be used to evaluate the calibration performance of low-cost $PM_{2.5}$ sensors by comparing its readings with hourly $PM_{2.5}$ measurement data of PA-II units.

Also, we updated the sentence on line 203 as follows:

These data suggest that a BAM-1020 instrument using non-FEM methods compares well to the statistics achieved with the FRM method.

10. Line 206 + 236: You list 6 significant figures/3 decimal points for several of the PA-II sensors, yet these sensors are not that accurate. As per the manufacturer +/-10 ug/m3 for 0-100 ug/m3 and +/-10% for 100-500 ug/m3. Please correct.

(Response)

We deleted Lines 206 and 235. Instead, we added the following footnote in Table 3 describing summary statistics of daily and hourly $PM_{2.5}$ concentrations from a FRM instrument, a BAM-1020 instrument, and a PA-II unit:

[Footnote] A PMS 5003 sensor that collects $PM_{2.5}$ concentrations from within a PA-II unit exhibits a maximum consistency error of +/-10 $\mu g/m^3$ at 0-100 $\mu g/m^3$ and +/-10% at 100-500 $\mu g/m^3$. The sensor reports $PM_{2.5}$ concentrations as integer values on a per-second basis. A PA-II unit generates readings of its own $PM_{2.5}$ concentrations by averaging its 1-second $PM_{2.5}$ concentrations over 80 (or 120) seconds. In this study, daily (hourly) $PM_{2.5}$ concentrations are calculated by averaging $PM_{2.5}$ concentrations rendered by a PA-II unit over 24 hours (1 hour), and thus can be represented with a decimal number. In other words, the presence of decimal

numbers in daily and hourly PM$_{2.5}$ concentrations reported by the PA-II 7 unit does not indicate precise concentration measurements.

**11. Line 219: How are you defining the r correlation of 0.928 as "good"?**

(Response)

We updated the sentence on line 209 as follows:

In this study, we examined the root mean square error (RMSE), mean squared error (MSE), mean absolute error (MAE), and Pearson correlation coefficient, $r$, between daily PM2.5 data from the FRM instrument and that from the PA-II units. In the cases of the RMSE, MSE, and MAE, the lower its value is, the better the performance or the lower the difference in measurement data between the FRM instrument and the PA-II units. The Pearson correlation coefficient is a metric measuring a linear correlation between two variables. It is a number between -1 and 1 that measures the strength and direction of their relationship. As the coefficient approaches an absolute value of 1, the values of measurement data from the FRM instrument and the PA-II units becomes more similar.

We updated the sentence on line 219 as follows:

These results show that the PA-II unit has a good correlation (r) with the FRM instrument for the two-year period of interest, since its value is very close to 1.

**12. Line 220: You say performance of FRM and BAM did not correlate favorably, yet in line 203 you state that the non-FEM method compared well to FRM? Why do you conclude that the BAM is less favorably correlated to the FRM when its statistics are better than the PAs?**

(Response)

We modified Line 220 as follows:

However, a comparison of metrics from the FRM instrument and the PA-II unit did not correlate as favorably.

13. Line 230: Please clarify why the FRM instrument was not used to evaluate hourly performance? Were hourly FRM measurements not available?

(Response)

We updated the sentence on line 233 as follows:

Next, we compared the PA-II unit's hourly $PM_{2.5}$ data with that of the BAM-1020 instrument over the course of the same two-year period. We did not consider the FRM instrument for exploring hourly $PM_{2.5}$ measurement data, since it only produces daily concentrations.

14. Line 272: The referenced article does not actually consider NO2 in their PM2.5 calibration. They only used PM2.5, Temperature, RH, CO, and wind speed in their models.

(Response)

The author of the article (Hua et al., 2021) claimed that $PM_{2.5}$ exhibits positive associations with $NO_2$, which indicates that $NO_2$ emissions make a large contribution to $PM_{2.5}$ pollution in the winter.

15. Line 293: "because month has a different slope..." Do you mean " because each month..."?

(Response)

We updated the sentence on line 293 as follows:

It is challenging to use the per-month linear fitting result to calibrate PA-II units because each month has a different slope and intercept defined for the linear fitting.

16. Lines 311 + 355: Can these lists be included as Tables rather than in-text to improve readability and when readers look at Tables 3-5.

(Response)

We added Tables for listing the selected feature vectors as a referee suggested.

17. Line 395: "Corresponding R2 values did not differ meaningfully" Based on what statistics, do you have a $p$-value?

(Response)

We updated the sentence on line 394 as follows:

For instance, the RMSE values from the best MLR and RF models were 3.912 $\mu g/m^3$ and 3.840 $\mu g/m^3$, respectively. Their corresponding $R^2$ values differ slightly, since their gap is only 0.008. Nonetheless, the MAE of 2.777 $\mu g/m^3$ achieved from the best MLR is lower than that achieved by the best RF, which is 2.831 $\mu g/m^3$.

18. Line 408: How are you defining moderate and high correlations?

(Response)

We updated the sentence on line 408 as follows:

The site 06-065-8005 had $NO_2$ measurements that are much more highly correlated with the Rubidoux site compared with those from the site 06-071-0027. This result can occur when the distance from the Rubidoux site to the site 06-065-8005 is shorter than it is to the site 06-071-0027.

19. Line 412: "We used NO2 for training a calibration model" Which NO2 data to train from, from Rubidoux? Please clarify.

(Response)

We rewrote Lines 410-413 as follows:

To evaluate the usefulness of distant $NO_2$ measurements on the calibration of a low-cost PM sensor, we used $NO_2$ data measured from monitoring sites near the PA-II 7 unit as a test dataset, rather than data from the collocated Rubidoux site. When we trained calibration models with the measurements from the PA-II 7 unit over 2018, we used highly accurate $NO_2$ concentrations

measured by FEM instruments at the Rubidoux site. Subsequently, to verify the trained calibration models, we utilized a separate test dataset featuring distant $NO_2$ measurements taken by FEM instruments at sites 06-065-8005 and 06-071-0027. We considered this scenario to evaluate our proposed calibration models, previously trained with collocated $NO_2$ concentrations and distant $NO_2$ concentrations, when collocated $NO_2$ measurements cannot be collected.

20. Line 430: "but not significantly" Based on what statistics, do you have a p-value?

(Response)

We updated the sentence on line 423 as follows:

All MLR methods using distant $NO_2$ data from site 06-071-0027 had a higher RMSE than the MLR algorithm was based on data that did not include $NO_2$ data from the collocated Rubidoux instrument, which had an RMSE of 4.513 $\mu g/m^3$ as shown in Table 5.

We updated the sentence on line 430 as follows:

In the case of RF models, the use of the distant $NO_2$ data from site 06-065-8005 increased RMSE compared to using collocated $NO_2$ data, but not significantly, since the maximum gap of RMSE values for all feature vectors considered was just 0.060 $\mu g/m^3$.

21. Line 447: Please re-word sentence as the point is unclear.

(Response)

We updated the sentence on line 447 as follows:

The factors, directly affecting the performance of a low-cost PM sensor, including temperature, relative humidity, and particle composition, have been scrutinized for their impact on sensors' performance enhancement.

22. Line 448: Please re-word to clarify that the inclusion of NO2 as an environmental factor in the calibration has potential to improve...

(Response)

We updated the sentence on line 448 as follows:

Additionally, this study investigated the potential of NO$_2$, a precursor gas that gives rise to PM$_{2.5}$ through atmospheric chemical reactions, to improve performance of the calibration model.

23. Section 2.2 Please include more information about the monitoring instrumentation used, especially the NO2 monitoring sites.

(Response)

We updated the sentence on line 110 as follows:

Monitoring ambient air quality for purposes of determining compliance with the U.S. National Ambient Air Quality Standards (NAAQSs) requires the use of either FRMs or FEMs. FRM and FEM instruments are accepted as methods for monitoring the NAAQS pollutants, including particulate matters (i.e., PM$_{2.5}$ and PM$_{10}$), NO$_2$, SO$_2$, O$_3$, and CO. Hourly measurements of PM$_{2.5}$, and other pollutants, such as NO$_2$, SO$_2$, O$_3$, and CO, obtained from FEM and non-FEM instruments can be downloaded via the EPA's application programming interface (https://aqs.epa.gov/data/api) (U.S. EPA, 2011).

{Reference}: U.S. EPA: Reference and Equivalent Method Applications: Guidelines for Applicants, Sep. 2011.

24. Section 3.2 + 3.3: At various points you include or drop units for your RMSE, MSE, MAE and r stats. Please be consistent. Shouldn't r (R2) be unitless? Please be consistent in using r vs R2.

(Response)

We modified these units throughout the paper, as recommended.

25. Section 3.6.3: Please check units of ug/m3 as you often have "ugm3" in this section.

(Response)

We modified these units throughout the paper, as recommended.

26. Equations 3, 4, & 5 could be included in the methods section rather than results.

(Response)

As recommended, we created a new subsection called Performance Evaluation Metrics, and moved the relevant paragraph to a new subsection.

---

## Author Comment (AC2)

**Response to Comments from Editors and Referees**

**Information of previously submitted manuscript**

- Manuscript number: egusphere-2023-1344
- Title: Evaluation of Calibration Performance of a Low-cost Particulate Matter Sensor Using Collocated and Distant $NO_2$
- Authors: Kabseok Ko, Ramesh R. Rao, and Seokheon Cho
- Status: Under Review

The authors would like to thank the referees for their careful reviews and valuable insight. We prepared our response to each of the editors' and referees' comments and revised our manuscript by reflecting all feedback.

**Referee #2**

The authors appreciate Referee #2's kind and valuable comments.

[Major Comments]

1. Very limited in scope and performance improvement: Given that the authors focused the comparison on calibration of a single sensor, the weight of "substantial contribution" of this manuscript falls on performance improvements of calibration models associated with that sensor. Unfortunately, the improvements on inclusion of NO2 are quite minimal. For example, in Tables 3 and 4, the best performances of models with and without NO2 are ~5% of each other. Does that qualify this work as "represent(ing) a substantial contribution to scientific progress" as is required by AMT? I disagree. I suggest that the authors conduct the analysis for the excluded sensor (sensor #8) that otherwise passes all checks, but was not included in the analysis for an unknown reason, as also pointed by reviewer 1.

(Response)

We appreciate the reviewer's perspective on the performance improvement for our proposed calibration models with the addition of $NO_2$ concentration as well as observation for one single PA-II unit. It may be that the improvements of around 5% observed in Tables 5 and 7 may not be substantial in absolute terms. However, it is important to consider the context and significance of these improvements.

First, even if calibration enhancement is modest in percentage, it can have practical implications in real-world application of low-cost $PM_{2.5}$ sensors, such as the PA-II units. A 5% improvement in low-cost $PM_{2.5}$ sensors can translate to more accurate and reliable measurements. The following three different feature vectors in Table 5 need to be addressed: feature vector #1, containing only $PM_{2.5;}$ feature vector #5 containing $PM_{2.5}$, temperature (T), and relative humidity (RH); and feature vector #16, consisting of a combination of $PM_{2.5}$, T, RH and $NO_2$.

We observed that the MLR-based calibration model considering feature vectors #1, #5, and #16 provides $R^2$ values of 0.731, 0.763, and 0.790, respectively. These values demonstrate that the MLR-based calibration model using $PM_{2.5}$, T, and RH results in an improvement of 4.4% in terms of $R^2$ compared to a calibration model only considering $PM_{2.5}$. Furthermore, the addition of $NO_2$ leads to an additional enhancement of 3.5% in comparison with the feature

vector consisting of $PM_{2.5}$, T, and RH. Hence, feature vector #16 can achieve a calibration performance improvement of up to 8.1% over feature vector #1, which uses only $PM_{2.5}$ concentrations. We must also consider that the PA-II units measuring $PM_{2.5}$ are low-cost sensors and may therefore face constraints in their performance. In other words, an $R^2$ of 0.790 is not easily the calibration model for a low-cost sensor. Hua *et al.* showed that a generalized additive model (GAM) using four variables, such as $PM_{2.5}$, T, RH, and CO, brings about a 7.3% improvement of $R^2$ compared to a GAM using one variable of $PM_{2.5}$ under dry conditions (Hua et al. 2021). Therefore, the authors consider an $R^2$ of 0.790 and the calibration improvement of 8.1% achieved by considering T, RH, and $NO_2$ to be significant results for calibration performance, especially taken across all four seasons.

Second, the significance of our study's contribution does not lie solely with the magnitude of performance improvement. The study's impact can also be evaluated in terms of its methodology, its novelty, and its potential to inspire further research. Including $NO_2$ measured by an expensive FEM-based device for calibration models, and not a collocated low-cost sensor, might be a novel approach that opens up new possibilities for research in this area.

Regarding the suggestion to conduct the analysis using PA-II 8 rather than the PA-II 7 unit used in original manuscript, it is a valid point raised by the reviewer 1 and thus we added analysis results for PA-II 8. We studied three cases of the PA-II 8 unit and showed that reliable and consistent PA-II units, which contain two PMS 5003 sensors with high correlation to each other, demonstrate similar calibration performance. This implies that a proposed calibration method can be applied to reliable and consistent PA-II units generally. The three case studies are included as follows:

Case 1: Calibration model is learned with the measurements collected from PA-II 8 in 2018 and calibration performance for the trained model is evaluated using data measured from PA-II 8 in 2019.

Case 2: This is similar to Case 1, except that the calibration model is trained with data measured from PA-II 7 in 2018.

Case 3: The measurement data from PA-II 8 with collocated NO2 concentration in 2018 is used as a training dataset, while the data collected from PA-II 8 with either collocated $NO_2$ or distant $NO_2$ concentration in 2019 is used as a test dataset.

2. Choice of performance parameters and lack of uncertainty analysis: While the authors include three performance measures, despite considering models with multiple and changing number of variables, the authors fail to include the most important one: adjusted R2. The authors have clearly used the multiple R2 squared value to compare model fits; however, multiple R2 will increase on addition of even poorly correlated variables. I suggest that the authors report adjusted R2 results. Additionally, presentation of such calibration results would also benefit from an uncertainty analysis, and a key manuscript cited by the authors uses bootstrapping to do just that (Hua et al., 2021). I strongly recommend uncertainty (in terms of standard deviation) be considered when presenting performance metrics associated with such comparisons. The authors can then answer the question: are the distributions of performance parameters statistically significantly different with or without NO2? I would consider answering that question as a significant contribution.

(Response)

We appreciate the reviewer's perspective about the performance metric $R^2$. The adjusted $R^2$ is formulated as follows:

$$adj\ R^2 = \frac{(N-1)R^2 - (M-1)}{N-M},$$

where N is the number of observations and M is the number of independent variables. To more accurately gauge the relationship between a PA-II unit and regulatory measurements over seasonality, we used whole-year data for training and test datasets, which are measured in 2018 and 2019, respectively. Our training and test datasets contained 7,198 and 7,621 samples, respectively. These numbers are much larger than the number of independent variables. Thus, from the equation of adjusted $R^2$ above, M has little effect on the value of adjusted $R^2$. In other words, adjusted $R^2$ is not significantly different from $R^2$ in our study. Table R1 shows the values of both $R^2$ and adjusted $R^2$ for an MLR-based calibration model on test datasets. The maximum difference between two values for every feature vector is 0.01.

**Table R1. Comparison of $R^2$ and adjusted $R^2$ for MLR-based calibration model.**

| Feature Vector | $R^2$ | Adjusted $R^2$ | Feature Vector | $R^2$ | Adjusted $R^2$ |
|---|---|---|---|---|---|
| 1 | 0.731 | 0.731 | 10 | 0.741 | 0.741 |
| 2 | 0.755 | 0.755 | 11 | 0.741 | 0.741 |
| 3 | 0.760 | 0.760 | 12 | 0.789 | 0.789 |
| 4 | 0.763 | 0.763 | 13 | 0.789 | 0.789 |

| 5 | 0.763 | 0.763 | 14 | 0.792 | 0.792 |
|---|-------|-------|----|-------|-------|
| 6 | 0.765 | 0.765 | 15 | 0.793 | 0.793 |
| 7 | 0.772 | 0.772 | 16 | 0.790 | 0.790 |
| 8 | 0.772 | 0.772 | 17 | 0.789 | 0.789 |
| 9 | 0.771 | 0.771 | 18 | 0.798 | 0.797 |
|   |       |       | 19 | 0.796 | 0.796 |
|   |       |       | 20 | 0.797 | 0.797 |
|   |       |       | 21 | 0.797 | 0.797 |

We performed an uncertainty analysis of the MLR-based calibration model by using a bootstrapping technique on a test dataset. Table R2 shows statistics of uncertainty analysis for each feature vector and t-values between two feature vectors whose difference is the existence of $NO_2$. We selected 8 feature vectors with various independent variables to verify whether the addition of $NO_2$ affects the performance of our calibration model. The 4 feature vectors we considered are $\{PM_{2.5}\}$, $\{PM_{2.5}, T\}$, $\{PM_{2.5}, RH\}$, and $\{PM_{2.5}, T, RH\}$. We also added $NO_2$ to create four other feature vectors, $\{PM_{2.5}, NO_2\}$, $\{PM_{2.5}, T, NO_2\}$, $\{PM_{2.5}, RH, NO_2\}$, and $\{PM_{2.5}, T, RH, NO_2\}$. We generated 1,000 test sets using a bootstrapping technique with replacement. We evaluated mean and standard deviation values of RSME calculated over 1,000 test sets for each feature vector. In addition, we applied a t-test to verify the effectiveness of adding $NO_2$ to each feature vector. Consideration of $NO_2$ additionally reduces mean values of RMSE for all 4 feature vectors. Contrary to mean value, standard deviation of RMSE for every feature vector increases slightly with the addition of $NO_2$.

We evaluated t-value for the mean values of RMSE for two feature vectors, with and without $NO_2$; for example, the t-value between $\{PM_{2.5}\}$ and $\{PM_{2.5}, NO_2\}$. Hence, we can evaluate 4 t-values. Degree of Freedom (DoF) is 1,998, so the relevant p-values are much less than 0.00001. Therefore, the difference in the mean RMSE values of the NO2–included and NO2-excluded groups is significant.

From these results, we can conclude that the performance of the MLR-based calibration model can be enhanced with consideration of $NO_2$ concentrations.

**Table R2. Statistics of uncertainty analysis to selected feature vectors and t-values.**

| Feature Vector | Mean of RMSE | Std. Dev. of RMSE | Feature Vector | Mean of RMSE | Std. Dev. of RMSE | t-value | DoF |
|----------------|--------------|-------------------|----------------|--------------|-------------------|---------|-----|
| $\{PM_{2.5}\}$ | 4.5095 | 0.1026 | $\{PM_{2.5}, NO_2\}$ | 4.4202 | 0.1037 | 19.3580 | 1,998 |

| | | | | | | |
|---|---|---|---|---|---|---|
| {$PM_{2.5}$, T} | 4.3084 | 0.1000 | {$PM_{2.5}$, T, $NO_2$} | 3.9979 | 0.1173 | 63.7008 | 1,998 |
| {$PM_{2.5}$, RH} | 4.2598 | 0.0995 | {$PM_{2.5}$, RH, $NO_2$} | 4.1548 | 0.1074 | 22.6792 | 1,998 |
| {$PM_{2.5}$, T, RH} | 4.2387 | 0.1050 | {$PM_{2.5}$, T, RH, $NO_2$} | 3.9865 | 0.1156 | 51.0686 | 1,998 |

3. Poor presentation: Large sections of the manuscript are unnecessarily detailed, and could be moved into tabular form whether in the main manuscript or the supplement. These include large portion of the lines 198-222 and 233-246 which are two representative examples. Additional examples include lists of variables shown in text format, which is laborious to read or keep track of (e.g., Lines 355-364). Additionally, key details of the authors' methodology such as performance metrics and intercomparison exercises are dispersed throughout the Results section (Sect. 3.1 to 3.6). I suggest that authors separate the methods portions of these results and discuss them in a separate subsection under Methods called "Instrument intercomparisons".

(Response)

We appreciate the reviewer's suggestion to streamline our presentation by consolidating certain portions into tabular format and creating a dedicated subsection within the Methods section. Hence, we implemented these changes to improve the overall clarity and accessibility of our work. We carefully reorganized the manuscript to enhance readability and ensure that we present key methodological details more cohesively.

1) We restructured Sections 2 and 3 as follows:
* * *
2. Methods

2.1 Measurement data

2.1.1 PurpleAir PA-II units

2.1.2 Air quality measurement data from EPA

2.1.3 Selection of PA-II units and reference monitoring sites

  - Note: We merged Subsection 2.3 with Subsection 3.1

2.1.4 Data preprocessing of PA-II units

2.2 Instrument intercomparisons

  - Note: We merged Subsections 3.2 and 3.3. We also eliminated redundancy by creating a table for summary statistics of daily and hourly $PM_{2.5}$ measurement data, and removing detailed explanations of maximum, minimum, mean, and standard deviations of various measurement data.
* * *
2.3 Feature vector selection for calibration models

   - We merged Subsections 3.3 and 3.4. We then shifted the merged text into this subsection and simplified the contents for greater cohesion from the viewpoint of feature selections.

2.4 Calibration models

2.4.1 Multiple Linear Regression (MLR)

2.4.2 Random Forest (FR)

2.5 Performance evaluation metrics

3. Results and discussion

3.1 Calibration performance

3.1.1 MLR-based calibration model

3.1.2 RF-based calibration model

3.2 Effect of distant $NO_2$ on calibration performance

3.3 Applicability of other PA-II Units

3.4 Effect of training period

3.5 Uncertainty analysis

   2) We added two Tables describing the selected feature vectors used in analyzing our MLR- and RF-based calibration models, which had previously been written as text in our original manuscript. In the original manuscript, this included Lines 311-318 and 354-362.

   3) We added a subsection on performance evaluation metrics to improve readability.

[Minor Comments]

1. Lines 123-139 The authors start off with a large dataset but remove data points using some filters. I suggest that the authors add a supplementary table showing how many data points were removed at each step.

(Response)

We included the following information in the supplementary document regarding the number of data points processed for each step of pre-processing.

**Table R1 Number of data points processed for each step of pre-processing.**

| Applied Method | Number of data points |
|---|---|
| Original (01/01/2018 – 12/31/2019) | 703,369 |
| Remove data with N/A | 703,369 |
| Valid data with 0<=Temperature<=200 | 703,368 |
| Valid data with 0<=RH<=100 | 703,339 |
| Valid data with $PM_{2.5}$ <= 2,000 | 703,339 |
| Averaging data for hourly $PM_{2.5}$ | 17,507 |
| Hourly Averaging with sufficient data points | 17,198 |
| Comparison of PMS 5003 A and B using SPE | 16,966 |

2. Lines 412-413 and Lines 287-290 The language used by the authors is unclear. I suggest either expanding on these sentences or rephrasing them so that the point is made clearly.

(Response)

We rewrote Lines 410-413 as follows:

To evaluate the usefulness of distant $NO_2$ measurements on the calibration of a low-cost PM sensor, we used $NO_2$ data measured from monitoring sites near the PA-II 7 unit as a test dataset, rather than data from the collocated Rubidoux site. When we trained calibration models with the measurements from the PA-II 7 unit over 2018, we used highly accurate $NO_2$ concentrations measured by FEM instruments at the Rubidoux site. Subsequently, to verify the trained calibration models, we utilized a separate test dataset featuring distant $NO_2$ measurements taken by FEM instruments at sites 06-065-8005 and 06-071-0027. We considered this scenario to evaluate our proposed calibration models, previously trained with collocated $NO_2$ concentrations and distant $NO_2$ concentrations, when collocated $NO_2$ measurements cannot be collected.

We rewrote Lines 287-290 as follows:

These remarkable results suggest that $NO_2$ is generally a key factor that can improve the performance of PA-II units over a year, even though the enhancement by $NO_2$ does not meet the values of 0.7 of $R^2$ and 3.5 $\mu g/m^3$ of RMSE during certain months, such as July 2018, August 2019, and October 2019.

---

## Author Comment (AC3)

**Response to Comments from Editors and Referees**

**Information of previously submitted manuscript**

- Manuscript number: egusphere-2023-1344
- Title: Evaluation of Calibration Performance of a Low-cost Particulate Matter Sensor Using Collocated and Distant $NO_2$
- Authors: Kabseok Ko, Ramesh R. Rao, and Seokheon Cho
- Status: Under Review

The authors would like to thank the referees for their careful reviews and valuable insights. We prepared our response to each of the editors' and referees' comments and revised our manuscript by reflecting all feedback.

**Referee #3**

The authors appreciate Referee #3's kind and valuable comments.

[Major Comments]

1. Unfortunately, the study structure and data do not support the authors' claims due to the lack of a robust dataset and an unclear strategy between model training and model evaluation data groups.

(Response)

Thank you for your kind and valuable comments. We acknowledge the need for a clearer representation of our strategy for training and testing calibration models within the paper's structure. To address this concern, we restructured the paper by emphasizing a more explicit delineation of our approach in training and testing the calibration models.

Specifically, we understand the potential for confusion with a per-month analysis based on MLR methods in the context of training and testing calibration models. The aim of monthly analysis was to illustrate the impact of $NO_2$ as a new feature vector on calibration performance in a monthly manner. To rectify this ambiguity, we have refined the structure by reorganizing the discussion of the monthly analysis method into the subsection "Feature Vector Selection for Calibration Models" within the Methods section.

2. The article is confusing and hard to follow. Too much detail is given for non-relevant information but not enough for evaluation.

(Response)

Thank you for your thorough feedback on the manuscript.

We streamlined the presentation by consolidating certain portions into a tabular format and creating a dedicated subsection, called "Instrument Intercomparisons," within the Methods section. We implemented these changes to improve the overall clarity and accessibility of our work.

We carefully reorganized the manuscript to enhance readability and ensure that we present key methodological details more cohesively.

1) We restructured Sections 2 and 3 as follows:

2. Methods

2.1 Measurement data

2.1.1 PurpleAir PA-II units

2.1.2 Air quality measurement data from EPA

2.1.3 Selection of PA-II units and reference monitoring sites

  - Note: We merged Subsection 2.3 with Subsection 3.1

2.1.4 Data preprocessing of PA-II units

2.2 Instrument intercomparisons

   - Note: We merged Subsections 3.2 and 3.3. We also eliminated redundancy by creating a table for summary statistics of daily and hourly $PM_{2.5}$ measurement data, and removing detailed explanations of maximum, minimum, mean, and standard deviations of various measurement data.

2.3 Feature vector selection for calibration models

   - We merged Subsections 3.3 and 3.4. We then shifted the merged text into this subsection and simplified the contents for greater cohesion from the viewpoint of feature selections.

2.4 Calibration models

2.4.1 Multiple Linear Regression (MLR)

2.4.2 Random Forest (FR)

2.5 Performance evaluation metrics

3. Results and discussion

3.1 Calibration performance

3.1.1 MLR-based calibration model

3.1.2 RF-based calibration model

3.2 Effect of distant $NO_2$ on calibration performance

2) We added two Tables describing the selected feature vectors used in analyzing our MLR- and RF-based calibration models, which had previously been written as text in our original manuscript. In the original manuscript, this included Lines 311-318 and 354-362.

3) We added a subsection on performance evaluation metrics to improve readability.

3. The authors argue against multivariable linear regression analyses but use MLR without offering a reasonable justification for its use nor explain why its results from RF and MLR are comparable.

(Response)

Thank you for your comment. We'd like to clarify our approach regarding the use of Multivariable Linear Regression (MLR) and Random Forest (RF) methods in our study.

We're not against MLR methods. Please refer to the following sentences in our manuscript: "A per-month analysis with a combination of features, including T, RH, and $NO_2$, showed an effect on calibration for the PA-II unit. It can be challenging to apply the per-month linear fitting result to calibrate PA-II units because month has a different slope and intercept defined for the linear fitting. Moreover, their values differ over two years even for the same month. For example, notably, the linear fitting result in Apr. 2018 exhibited a higher RMSE than the linear fitting result yielded in Apr. 2019. On the contrary, the calibration performance in Aug. 2018 was worse than that in Aug. 2019." Here, we are saying that we don't think performing MLR-based calibrations on a monthly basis is a good approach to proofreading. In addition, the monthly MLR analyses were primarily conducted for feature selection rather than calibration, specifically to confirm the viability of $NO_2$ as a feature for improving calibration performance. In evaluating the impact of $NO_2$, we considered MLR and RF algorithms.

Our findings revealed a significant enhancement in the calibration performance of both MLR and RF models after incorporating $NO_2$ concentrations. This inclusion notably reduced the performance disparity between MLR and RF models, which resulted in enhancement of calibration performance for both methodologies.

[Minor Comments]

1. Title misspelled "Collocated".

(Response)
We modified "colocated" to "collocated".

2. Line 47: "however" seems to be misplaced.

(Response)
We updated as follows:
However, low-cost PM sensors are not suitable for regulatory purposes because the data reported can be questionable in terms of accuracy, precision, and reliability.

3. Line 121: This sentence is poorly constructed and confusing.

(Response)
We rewrote the sentence as follows:
Therefore, PA-II units may have abnormal data due to failure and aging drift, so data quality control is required before calibrating the PA-II units.

---

## Author Response (AR2)

**Response to Editor's and Reviewers' Comments**

**Information of previously submitted manuscript**

- Manuscript number: egusphere-2023-1344
- Title: Evaluation of Calibration Performance of a Low-cost Particulate Matter Sensor Using Colocated and Distant $NO_2$
- Authors: Kabseok Ko, Seokheon Cho, and Ramesh R. Rao
- Status: Minor Revision

The authors would like to thank the referees for their careful reviews and insightful comments. We prepared responses for each comment from the editors and referees and revised our manuscript to reflect feedback.

The authors appreciate Reviewer #1's kind and valuable comments.

[Major Comments]

1. Your article idea is interesting and relevant. However, the data set is very small (two PA-II units, when only one is used in most of the analysis). Would it be possible to add other locations/states to make your method validation more robust?

(Response)

We sought to evaluate the applicability of our proposed methods across diverse locations. We used to download the PM2.5 concentrations from the open platform operated by PurpleAir Inc. that manufactures the PA-II units. However, we encountered their recent policy change, which now imposes charges for data downloads and then makes us challenging to obtain a sufficient volume of data.

Hence, we searched for the data we had downloaded from PurpleAir open platform. Fortunately, we identified one single suitable PA-II unit that meets our criteria, which a PA-II unit is collocated to an EPA monitor measuring both $PM_{2.5}$ and $NO_2$ (used as collocated $NO_2$), as well as there exists at least other EPA monitor measuring $NO_2$ (used as nearby $NO_2$) not being far away from the collocated EAP monitoring station. This particular PA-II unit, named SACRAMENTO, is located in Sacramento, CA, USA. It is installed near an EPA monitoring site, designated as 06-067-0010. Another EPA monitoring site to measure $NO_2$, 06-067-0015, is located in the vicinity of the collocated EAP station.

We found around 13 months of data from December 2020 to December 2021 that we had collected from the PA-II named SACRAMENTO earlier. In our manuscript, to utilize and scrutinize the seasonality patterns of $PM_{2.5}$ concentrations, we divided the two-year dataset into a training dataset and test dataset consisting of data collected in 2018 and 2019, respectively. We cannot apply our previous approach onto the SACRAMENTO PA-II unit due to the limited data we had stored. However, we could keep considering seasonality for reliable performance by employing stratified random sampling to partition the dataset. The stratified strategy was applied on a monthly basis with the split ratio of 80:20 for training and testing. We assessed

the calibration performance of the SACRAMENTO PA-II unit using the MLR algorithm and the corresponding results are presented in Table R1.

When $NO_2$ is not included in feature vectors, the best performance is achieved for feature vector #9. Also, in case of using NO2, all feature combination sets from #12 to #20 except #15 result in higher $R^2$ as well as lower RMSE and MAE than feature vector #9. This is the same result as we showed with two PA-II units in our manuscript.

**Table R1. Calibration results of hourly PM$_{2.5}$ concentrations measured from the SACRAMENTO PA-II using MLR-based calibration model.**

| NO$_2$ not included | | | | NO$_2$ included | | | |
|---|---|---|---|---|---|---|---|
| Feature Vector | $R^2$ | RMSE | MAE | Feature Vector | $R^2$ | RMSE | MAE |
| 1 | 0.743 | 4.844 | 3.070 | 10 | 0.745 | 4.824 | 3.018 |
| 2 | 0.764 | 4.646 | 3.032 | 11 | 0.773 | 4.552 | 2.909 |
| 3 | 0.769 | 4.597 | 2.982 | 12 | 0.790 | 4.385 | 2.790 |
| 4 | 0.783 | 4.458 | 2.874 | 13 | 0.791 | 4.372 | 2.767 |
| 5 | 0.769 | 4.590 | 2.986 | 14 | 0.791 | 4.371 | 2.768 |
| 6 | 0.776 | 4.520 | 2.943 | 15 | 0.776 | 4.527 | 2.903 |
| 7 | 0.783 | 4.452 | 2.875 | 16 | 0.791 | 4.371 | 2.787 |
| 8 | 0.784 | 4.441 | 2.859 | 17 | 0.792 | 4.359 | 2.767 |
| 9 | 0.784 | 4.441 | 2.857 | 18 | 0.792 | 4.357 | 2.767 |
| | | | | 19 | 0.793 | 4.346 | 2.757 |
| | | | | 20 | 0.793 | 4.354 | 2.765 |

We tried to evaluate the performance of MLR-based calibration model to get the comparison results using both collocated $NO_2$ (06-067-0010) and distant $NO_2$ (06-067-0015) measurements. However, the site we need to refer to as distant $NO_2$ has discontinued measuring $NO_2$ since Aug. 2021. Hence, we can gather and use only around 7 months of $NO_2$ data, which precludes a fair and appropriate performance comparison between collocated $NO_2$ and distant $NO_2$ as we provided in our manuscript.

2. Line 77: check spelling of "gase".

(Response)

We had already used "gases" on Line 77 as you suggested, so we don't change the word.

3. Sections 2.1.2, 2.1.3, and 2.1.4:
- What is the relevance of explaining POC and other EPA minutia?.
- Why give so much attention to sections 2.1.2 and 2.1.3 but not detail the test and training data separation strategy on 2.1.4? - - I recommend using the level of detail used in section 2.2

(Response)

We modified on line 103 (removed sentences from line 104 to line 112) in subsection 2.1.2 as follows:

Outdoor air quality data collected from across the U.S. is publicly available through the U.S. Environmental Protection Agency (EPA) website (https://epa.gov/outdoor-air-quality-data). ~~The EPA has a description file for monitors, which includes state code, county code, site number, location (latitude and longitude), parameter code, parameter occurrence code (POC), and last method. A combination of state code, county code, and site number can uniquely identify a monitoring site. For example, a monitoring station located at Bakersfield, CA has a state code of 06, a county code of 029, and a site number of 0014. The parameter code is an air quality system (AQS) code corresponding to the parameter measured by a monitor. For example, parameters regarding PM$_{2.5}$ and NO$_2$ are 88101 and 42602, respectively. A POC is used to identify an instrument among multiple ones with the same parameter code at a site. For example, two FRM instruments with a parameter of 88101 at the Bakersfield site are used to measure daily PM$_{2.5}$ concentrations and are identified with POC 1 and 2. The last method descriptor describes the measurement scheme used by the monitor for its most recent sample.~~

We added a new paragraph about test and training data separation strategy after line 170 in subsection 2.1.4 of the original manuscript as follows:

The period of valid measurement data collected from the PA-II units we selected is 24 months, such as from Jan. 2018 to Dec. 2019. The measurement data in the years 2018 and 2019 from the two-year dataset were used for training and testing for our calibration models, respectively. The reason why we split the two-year dataset at a 1:1 ratio is that PM2.5 as well as the other environmental parameters, such as temperature and relative humidity, which we considered for calibration models, have a seasonal pattern. Also, we used whole-year dataset for training to learn the relationship between PA-II and regulatory measurement over seasonality and thus enhance the performance of the calibration models over all 4 seasons.

4. Line 152: "...measure 'for' obtaining..."? Are you sure it is 'for' and not 'by'?

(Response)

We changed "for" to "by".

**Reviewer: 2**

The authors appreciate Reviewer #2's kind and valuable comments.

[Minor Comments] Comments to the Corresponding Author

1. Line 62: Please change to "two months have shown good correlation"

(Response)

We corrected "have shown" as you suggested.

2. Line 322, 367: Pleas unitalicize ug/m^3

(Response)

We changed italic ug/m^3 into unitalic ug/m^3.